# Fast Training Methods for Stochastic Compositional Optimization Problems

**Hongchang Gao**[1]**, Heng Huang**[2]

[1] Department of Computer and Information Sciences, Temple University, PA, USA

[2] Department of Electrical and Computer Engineering, University of Pittsburgh, PA, USA
hongchang.gao@temple.edu, heng.huang@pitt.edu

## Abstract

The stochastic compositional optimization problem covers a wide range of machine learning models, such as sparse additive models and model-agnostic meta-learning. Thus, it is necessary to develop efficient methods for its optimization. Existing methods for the stochastic compositional optimization problem only focus on the single machine scenario, which is far from satisfactory when data are distributed on different devices. To address this problem, we propose novel decentralized stochastic compositional gradient descent methods to efficiently train the large-scale stochastic compositional optimization problem. To the best of our knowledge, our work is the first one facilitating decentralized training for this kind of problem. Furthermore, we provide the convergence analysis for our methods, which shows that the convergence rate of our methods can achieve linear speedup with respect to the number of devices. At last, we apply our decentralized training methods to the model-agnostic meta-learning problem, and the experimental results confirm the superior performance of our methods.

## 1  Introduction

The stochastic compositional optimization problem [20] plays an important role in the machine learning field, since it covers numerous applications, such as policy evaluation [19], sparse additive models [20], and model-agnostic meta-learning [6, 1]. Specifically, the stochastic compositional optimization problem is defined as follows:

$$\min_{x \in \mathbb{R}^d} \mathbb{E}_\zeta \left[ f \left( \mathbb{E}_\xi [g(x; \xi)]; \zeta \right) \right], \tag{1}$$

where the outer-level function $f(y) = \mathbb{E}_\zeta[f(y; \zeta)] : \mathbb{R}^{d'} \to \mathbb{R}$ is a smooth and nonconvex function, the inner-level function $g(x) = \mathbb{E}_\xi[g(x; \xi)] : \mathbb{R}^d \to \mathbb{R}^{d'}$ is a smooth function. It is obvious that the loss function consists of two stochastic functions $f$ and $g$, making it different from the non-compositional problem.

The compositional structure in Eq. (1) leads to more challenges in optimization compared with the non-compositional problem, since the stochastic gradient of the loss function in Eq. (1) is not an unbiased estimation of the full gradient. Thus, efficient training of the stochastic compositional optimization problem has attracted increasing attention in recent years. For example, [20] proposed the stochastic compositional gradient descent (SCGD) method to deal with the two-level stochasticity in Eq. (1). But it has a worse convergence rate than the standard stochastic gradient descent method. To improve it, a series of variance-reduced methods have been proposed. For example, [25] developed the composite randomized incremental gradient method based on the SAGA [4] technique. [27, 24] further improved the convergence rate based on the SPIDER [5] method.

35th Conference on Neural Information Processing Systems (NeurIPS 2021).

Although aforementioned methods have achieved much progress in optimizing Eq. (1), all of them only focus on the single-machine scenario. In fact, data are usually distributed on different devices in many real-world applications. It is necessary to study the distributed training method for the stochastic compositional optimization problem. In the past few years, numerous distributed training methods have been proposed, such as asynchronous stochastic gradient descent (SGD) [16], decentralized SGD [13], local SGD [14]. But they are not applicable to the stochastic compositional problem. Especially, their convergence results do not hold for Eq. (1) because those distributed training methods require the stochastic gradient is an unbiased estimation of the full gradient. As far as we know, there are no existing works studying distributed training methods for Eq. (1) with theoretical guarantees.

To fill the aforementioned gap, we developed two novel decentralized training methods for the stochastic compositional optimization problem. In particular, we proposed a *gossip-based* decentralized stochastic compositional gradient descent (GP-SCGD) method and a *gradient-tracking-based* decentralized stochastic compositional gradient descent (GT-SCGD) method. With our methods, different devices can collaboratively optimize Eq. (1). However, establishing the convergence rate of our proposed methods is challenging. Existing convergence analysis techniques for decentralized SGD are not applicable to our methods because the *stochastic compositional gradient* is different from the *stochastic gradient*. To address this challenge, we proposed new convergence analysis techniques for the decentralized SCGD method, where we show how to bound the gradient variance and the consensus error under the decentralized setting. In particular, both GP-SCGD and GT-SCGD can achieve the convergence rate $O(\frac{1}{K\epsilon^2})$ to achieve $\epsilon$-accuracy solution, where $K$ is the number of devices. It indicates that our methods can achieve the linear speedup with respect to the number of devices, which is consistent with decentralized SGD. To the best of our knowledge, our work is the first one to establish the convergence rate of decentralized SCGD and achieving the linear speedup. Moreover, the sample complexity of our methods is $O(\frac{1}{K\epsilon^3})$. It indicates that our methods can achieve a better sample complexity $O(\frac{1}{\epsilon^3})$ than $O(\frac{1}{\epsilon^4})$ of traditional SCGD [20] when $K = 1$, which further confirms the superiority of our methods. Finally, we applied our proposed methods to optimize the model-agnostic meta-learning problem. The empirical results confirm the effectiveness of our methods. In the following, we summarize the contributions of our work.

- We proposed two novel decentralized stochastic compositional gradient descent methods: GP-SCGD and GT-SCGD. This is the first work studying the decentralized training method for stochastic compositional optimization problems.

- We established the convergence rate of our decentralized stochastic compositional gradient descent methods with novel theoretical analysis. This is the first work establishing the convergence rate for decentralized SCGD and showing the linear speedup with respect to the number of devices. In addition, our methods can achieve better sample complexities than traditional SCGD.

- The extensive empirical evaluation on the model-agnostic meta-learning task confirms the effectiveness of our proposed methods.

## 2 Related Works

### 2.1 Stochastic Compositional Optimization Problem

The stochastic compositional optimization problem is common in the machine learning field. However, it is more difficult to optimize than the standard non-compositional problem. The reason is that its stochastic gradient is not an unbiased estimation of the full gradient, which is shown as follows:

$$\mathbb{E}_{\xi,\zeta}[\nabla g(x;\xi)^T \nabla_g f(g(x;\xi);\zeta)] \neq \nabla g(x)^T \nabla_g f(g(x)) . \tag{2}$$

To train the stochastic compositional problem efficiently, numerous methods have been proposed in the past few years. Specifically, [20] developed the stochastic compositional gradient descent (SCGD) method, which is defined as follows:

$$
\begin{aligned}
u_t &= (1 - \beta_{t-1})u_{t-1} + \beta_{t-1}g(x_t; \mathcal{B}_{\xi,t}) , \\
x_{t+1} &= x_t - \eta_t \nabla g(x_t; \mathcal{B}_{\xi,t})^T \nabla_g f(u_t; \mathcal{A}_{\zeta,t}) ,
\end{aligned}
\tag{3}
$$

where $\mathcal{B}_{\xi,t}$ and $\mathcal{A}_{\zeta,t}$ denote the random samples, $0 < \beta_t < 1$ is a hyperparameter, and $\eta_t$ is the learning rate. Here, $u_t$ serves as the estimator of the inner function $g(x_t)$, and then the gradient

$\nabla_g f(g(x_t))$ is estimated by $\nabla_g f(u_t; \mathcal{A}_{\zeta,t})$. As a result, this stochastic compositional gradient $\nabla g(x_t; \mathcal{B}_{\xi,t})^T \nabla_g f(u_t; \mathcal{A}_{\zeta,t})$ has a smaller variance to improve the accelerate the convergence speed. However, it still has a worse theoretical convergence rate than SGD. Furthermore, to improve the convergence speed of SCGD for nonconvex problems, advanced variance reduction techniques have been incorporated into SCGD. For example, [25] incorporated the variance reduction technique SAGA [15] to SCGD and obtained a better convergence rate. Later, [27] employed another variance reduction technique SPIDER [5] for SCGD, while [23] accelerated SCGD with the STORM [3] variance reduction technique. More recently, some works [2, 10] disclosed the connection between stochastic bilevel optimization problems and stochastic compositional problems and developed the alternating SGD method to solve Eq. (1). However, all of these methods only studied the convergence rate for the single machine case, ignoring the distributed setting.

## 2.2 Decentralized Training Methods

Recently, with the emergence of large-scale data, numerous distributed training methods [16, 13, 14, 18, 11, 7, 9, 8] have been proposed. Among them, the decentralized training method, where different devices conduct the peer-to-peer communication, has attracted a lot of attention. In particular, there are two kinds of communication strategies in a decentralized training system. They are the *gossip* strategy and the *gradient tracking* strategy. Based on the gossip strategy, [13] studied the convergence rate of decentralized SGD and disclosed how the topology of the communication graph affects the convergence rate. Additionally, numerous works aim to reduce the communication cost of the gossip-based decentralized SGD by compressing gradients or skipping the communication step. Meanwhile, based on the gradient-tracking strategy, [17] developed the distributed stochastic gradient tracking method and established its convergence rate. Recently, some works [18, 22] combined the variance reduction strategy and the gradient-tracking strategy to further accelerate decentralized SGD. However, all of these methods are not applicable to SCGD. Thus, in this paper, we will develop efficient decentralized SCGD methods for optimizing the stochastic compositional optimization problem.

# 3 Decentralized Stochastic Compositional Optimization Methods

## 3.1 Problem Definition

Formally, we consider the distributed stochastic compositional optimization problem, which is defined as follows:

$$\min_{x \in \mathbb{R}^d} \frac{1}{K} \sum_{k=1}^{K} \mathbb{E}_\zeta \left[ f^{(k)} \left( \mathbb{E}_\xi [g^{(k)}(x;\xi)]; \zeta \right) \right], \tag{4}$$

where $x \in \mathbb{R}^d$ denotes the model parameter, $K$ is the total number of devices, $f^{(k)} = \mathbb{E}_\zeta[f^{(k)}(y;\zeta)] : \mathbb{R}^{d'} \to \mathbb{R}$ and $g^{(k)} = \mathbb{E}_\xi[g^{(k)}(x;\xi)] : \mathbb{R}^d \to \mathbb{R}^{d'}$ are smooth functions on the $k$-th device, and accordingly $\mathbb{E}_\zeta \left[ f^{(k)} \left( \mathbb{E}_\xi [g^{(k)}(x;\xi)]; \zeta \right) \right]$ is the loss function on the $k$-th device. In other words, each device has its own data and all devices collaboratively learn the model parameter $x$ by utilizing their local data.

To efficiently train this distributed stochastic compositional optimization problem, we consider the decentralized training method. In particular, each device connects with its neighbors, composing a communication network. All devices conduct the peer-to-peer communication based on this communication network. Formally, we represent the communication network with $\mathcal{G} = (V, W)$ where $V = \{v_1, v_2, \cdots, v_K\}$ denotes all devices and $W = [w_{ij}] \in \mathbb{R}^{K \times K}$ is the adjacency matrix which indicates whether two devices are connected or not. Following [11, 13], the adjacency matrix $W$ is assumed to satisfy the following assumption.

**Assumption 1.** *The adjacency matrix $W$ has the following properties:*

- *Nonnegative: $w_{ij} \geq 0$, $\forall i, j$.*

- *Symmetric: $w_{ij} = w_{ji}$, $\forall i, j$.*

- *Doubly Stochastic: $\sum_{i=1}^{K} w_{ij} = 1$, $\sum_{j=1}^{K} w_{ij} = 1$.*

- *The eigenvalues of $W$ can be sorted as $|\lambda_n| \leq \cdots \leq |\lambda_2| < |\lambda_1| = 1$.*

In our convergence analysis, we denote $\lambda = |\lambda_2|$ for simplicity. In terms of the aforementioned decentralized training setting, we proposed two descentralized stochastic compositional gradient descent methods. The details are described in the following subsection.

## 3.2 Gossip-based Decentralized Stochastic Compositional Gradient Descent Method

In Algorithm 1, we developed the *gossip*-based decentralized stochastic compositional gradient descent (GP-DSCGD) method. Following the standard SCGD method [20], each device estimates the inner-level function $g^{(k)}(x_t^{(k)})$ with $u_t^{(k)}$ as follows:

$$u_t^{(k)} = (1 - \gamma\beta_{t-1})u_{t-1}^{(k)} + \gamma\beta_{t-1}g^{(k)}(x_t^{(k)}; \mathcal{B}_t^{(k)}) \,, \tag{5}$$

where $\gamma > 0$ and $\beta_t > 0$ are two hyperparameters, $\gamma\beta_t < 1$, $k$ is the index of devices and $t$ is the index of iterations. Different from the standard SCGD method in Eq. (3), there is an additional hyperparameter $\gamma$ when computing $u_t^{(k)}$, which can help controlling the estimation variance. We will show it in our theoretical analysis in Appendix. Based on this estimation, each device computes the stochastic compositional gradient $z_t^{(k)}$ as shown in Line 9. Then, each worker uses the gossip communication strategy to update the local model parameter as follows:

$$\tilde{x}_{t+1}^{(k)} = \sum_{j \in \mathcal{N}_{v_k}} w_{kj}x_t^{(j)} - \eta z_t^{(k)} \,, \tag{6}$$

where $\eta > 0$ denotes the learning rate, $\mathcal{N}_{v_k} = \{j | w_{kj} > 0\}$ is the neighboring devices of the $k$-th device. Here, the first term on the right-hand side incurs the communication to average the model parameter of neighboring devices, and the second term indicates to update the model parameter with the local gradient $z_t^{(k)}$. Then, instead of using $\tilde{x}_{t+1}^{(k)}$ as the new model parameter, we compute the model parameter of the $(t+1)$-th iteration as follows:

$$x_{t+1}^{(k)} = x_t^{(k)} + \beta_t(\tilde{x}_{t+1}^{(k)} - x_t^{(k)}). \tag{7}$$

In fact, this strategy couples the update of $u_t^{(k)}$ and $x_t^{(k)}$ by the sharing hyperparameter $\beta_t$, which can benefit controlling the estimation variance of $u_t^{(k)}$. All devices repeat the aforementioned steps until the iterates converge.

---

**Algorithm 1** Gossip-based Decentralized Stochastic Compositional Gradient Descent (GP-DSCGD)

---

**Input:** $x_0^{(k)} = x_0$, $\beta_t > 0$, $\gamma > 0$, $\eta > 0$.
1: **for** $t = 0, \cdots, T-1$, each device $k$ **do**
2:    Sample a subset of samples $\mathcal{B}_t^{(k)}$ to compute:
3:    **if** $t = 0$ **then**
4:       $u_t^{(k)} = g^{(k)}(x_t^{(k)}; \mathcal{B}_t^{(k)})$
5:    **else**
6:       $u_t^{(k)} = (1 - \gamma\beta_{t-1})u_{t-1}^{(k)} + \gamma\beta_{t-1}g^{(k)}(x_t^{(k)}; \mathcal{B}_t^{(k)})$
7:    **end if**
8:    $v_t^{(k)} = \nabla g^{(k)}(x_t^{(k)}; \mathcal{B}_t^{(k)})$
9:    Sample a subset of samples $\mathcal{A}_t^{(k)}$ to compute:
      $z_t^{(k)} = (v_t^{(k)})^T \nabla f^{(k)}(u_t^{(k)}; \mathcal{A}_t^{(k)})$
10:    $\tilde{x}_{t+1}^{(k)} = \sum_{j \in \mathcal{N}_{v_k}} w_{kj}x_t^{(j)} - \eta z_t^{(k)}$
11:    $x_{t+1}^{(k)} = x_t^{(k)} + \beta_t(\tilde{x}_{t+1}^{(k)} - x_t^{(k)})$
12: **end for**

---

## 3.3 Gradient-Tracking-based Decentralized Stochastic Compositional Gradient Descent Method

Besides the gossip-based communication strategy, the gradient tracking strategy is also widely used in the decentralized training method. Therefore, for completeness, we further proposed the gradient-tracking-based descentralized stochastic compositional gradient descent (GT-DSCGD) method in

Algorithm 2. Same with Algorithm 1, we compute $u_t^{(k)}$ and $z_t^{(k)}$ as shown in Lines 2-9. Different from Algorithm 1, GT-DSCGD introduces an auxiliary variable $s_t^{(k)}$ for each device, which is used to track the global stochastic compositional gradient $\frac{1}{K}\sum_{k=1}^K z_t^{(k)}$ [18, 17]. Specifically, $s_t^{(k)}$ is updated as follows:

$$s_t^{(k)} = \sum_{j \in \mathcal{N}_{v_k}} w_{kj} s_{t-1}^{(j)} + z_t^{(k)} - z_{t-1}^{(k)} . \tag{8}$$

It is easy to verify that $\frac{1}{K}\sum_{k=1}^K s_t^{(k)} = \frac{1}{K}\sum_{k=1}^K z_t^{(k)}$. Then, each device uses $s_t^{(k)}$ instead of $z_t^{(k)}$ to update the model parameter as follows:

$$\tilde{x}_{t+1}^{(k)} = \sum_{j \in \mathcal{N}_{v_k}} w_{kj} x_t^{(j)} - \eta s_t^{(k)} . \tag{9}$$

To sum up, the difference between our two methods is that GT-DSCGD has an additional step to track the gradient with $s_t^{(k)}$ and accordingly updates the model parameter with $s_t^{(k)}$.

---

**Algorithm 2** Gradient-Tracking-based Decentralized Stochastic Compositional Gradient Descent (GT-DSCGD)

**Input:** $x_0^{(k)} = x_0$, $z_{-1}^{(k)} = 0$, $s_{-1}^{(k)} = 0$, $\beta_t > 0$, $\gamma > 0$, $\eta > 0$.
1: **for** $t = 0, \cdots, T-1$ , each device $k$ **do**
2:     Sample a subset of samples $\mathcal{B}_t^{(k)}$ to compute:
3:     **if** $t = 0$ **then**
4:        $u_t^{(k)} = g^{(k)}(x_t^{(k)}; \mathcal{B}_t^{(k)})$
5:     **else**
6:        $u_t^{(k)} = (1 - \gamma\beta_{t-1}) u_{t-1}^{(k)} + \gamma\beta_{t-1} g^{(k)}(x_t^{(k)}; \mathcal{B}_t^{(k)})$
7:     **end if**
8:     $v_t^{(k)} = \nabla g^{(k)}(x_t^{(k)}; \mathcal{B}_t^{(k)})$
9:     Sample a subset of samples $\mathcal{A}_t^{(k)}$ to compute:
       $z_t^{(k)} = (v_t^{(k)})^T \nabla f^{(k)}(u_t^{(k)}; \mathcal{A}_t^{(k)})$
10:    $s_t^{(k)} = \sum_{j \in \mathcal{N}_{v_k}} w_{kj} s_{t-1}^{(j)} + z_t^{(k)} - z_{t-1}^{(k)}$ // gradient tracking
11:    $\tilde{x}_{t+1}^{(k)} = \sum_{j \in \mathcal{N}_k} w_{kj} x_t^{(j)} - \eta s_t^{(k)}$
12:    $x_{t+1}^{(k)} = x_t^{(k)} + \beta_t(\tilde{x}_{t+1}^{(k)} - x_t^{(k)})$
13: **end for**

---

## 4 Convergence Analysis

### 4.1 Assumptions

To establish the convergence rate of our proposed two methods, we first introduce some standard assumptions, which are also used in existing stochastic compositional gradient descent methods [20, 21, 26, 24].

**Assumption 2.** *(Smoothness) For $\forall y_1, y_2 \in \mathbb{R}^{d'}$ and $\forall x_1, x_2 \in \mathbb{R}^d$, there exist two constant values $L_f > 0$ and $L_g > 0$ such that*

$$\|\nabla f^{(k)}(y_1) - \nabla f^{(k)}(y_2)\| \le L_f \|y_1 - y_2\|, \ \|\nabla g^{(k)}(x_1) - \nabla g^{(k)}(x_2)\| \le L_g \|x_1 - x_2\| . \tag{10}$$

**Assumption 3.** *(Bounded gradient) For $\forall x \in \mathbb{R}^d$ and $\forall y \in \mathbb{R}^{d'}$, there exist two constant values $C_g > 0$ and $C_f > 0$ such that*

$$\mathbb{E}[\|\nabla g^{(k)}(x; \xi)\|^2] \le C_g^2, \ \mathbb{E}[\|\nabla f^{(k)}(y; \zeta)\|^2] \le C_f^2 . \tag{11}$$

**Assumption 4.** *(Bounded variance) For $x \in \mathbb{R}^d$ and $y \in \mathbb{R}^{d'}$, there exist three constant values $\sigma_f > 0$, $\sigma_g > 0$, $\sigma_{g'} > 0$, such that*

$$\mathbb{E}[\|\nabla f^{(k)}(y; \zeta) - \nabla f^{(k)}(y)\|^2] \le \sigma_f^2, \ \mathbb{E}[\|\nabla g^{(k)}(x; \xi) - \nabla g^{(k)}(x)\|^2] \le \sigma_{g'}^2,$$
$$\mathbb{E}[\|g^{(k)}(x; \xi) - g^{(k)}(x)\|^2] \le \sigma_g^2 . \tag{12}$$

To study the convergence rate of our two methods, we denote $F(x) = \frac{1}{K}\sum_{k=1}^{K} F^{(k)}(x) = \frac{1}{K}\sum_{k=1}^{K} f^{(k)}(g^{(k)}(x))$. Then, $F(x)$ is $L_F$-smooth with $L_F = C_g^2 L_f + C_f L_g$. In addition, we denote $\bar{x}_t = \frac{1}{K}\sum_{k=1}^{K} x_t^{(k)}$ and $x_*$ as the optimal solution.

## 4.2 Convergence Rate

In terms of aforementioned definitions and assumptions, we establish convergence rates of our proposed Algorithm 1 and Algorithm 2, respectively.

**Theorem 1.** *Assume Assumptions 1-4 hold, for Algorithm 1, by setting $|\mathcal{A}_t^k| = |\mathcal{B}_t^k| = B$, $\gamma > 0$, $\beta_t = \beta \leq \min\{\frac{1}{8\gamma}, \frac{1}{2\eta L_F}, 1\}$, $\eta \leq \min\{\eta_1, \eta_2, \eta_3\}$ where*

$$\eta_1 = \frac{\gamma}{3\sqrt{6}C_g^2 L_f}, \eta_2 = \frac{1-\lambda^2}{2}\Big/\Big(\frac{27C_g^4 L_f^2 + \gamma^2 L_F^2}{\gamma^2} + \frac{\gamma(24C_f^2 L_g^2 + 99C_g^4 L_f^2)}{2C_g^2 L_f^2}\Big),$$

$$\eta_3 = \frac{\sqrt{b^2+4ac}-b}{2a}, a = \frac{(24C_f^2 L_g^2 + 3C_g^4 L_f^2)}{8(1-\lambda^2)C_g^2 L_f^2} + \frac{12C_g^2}{1-\lambda^2} + \frac{27C_g^4 L_f^2}{2\gamma^2}, b = \frac{2}{1-\lambda^2}, c = \frac{\gamma}{4C_g^2 L_f^2},$$

$$(13)$$

*we can get*

$$\frac{1}{T}\sum_{t=0}^{T-1}\mathbb{E}[\|\nabla F(\bar{x}_t)\|^2] \leq \frac{2\mathbb{E}[F(x_0) - F(x_*)]}{\eta\beta T} + \frac{2(3C_g^2\sigma_f^2 + 3C_f^2\sigma_{g'}^2 + C_g^2 L_f^2\sigma_g^2 + 24\gamma\sigma_g^2)}{B}$$

$$+ \frac{48C_f^2\sigma_{g'}^2 + 16C_g^2\sigma_f^2 + 99C_g^2 L_f^2\sigma_g^2}{16\beta C_g^2 L_f^2 B} + \frac{6C_g^2 L_f^2\sigma_g^2}{\beta\gamma B} + \frac{48\sigma^2}{\beta B}.$$

$$(14)$$

**Corollary 1.** *Assume Assumptions 1-4 hold, for Algorithm 1, by setting $\eta = O(\frac{\sqrt{K}}{\sqrt{T}})$, $B = O(\sqrt{KT})$, $\beta$ and $\gamma$ to be two positive constant values, we can get*

$$\frac{1}{T}\sum_{t=0}^{T-1}\mathbb{E}[\|\nabla F(\bar{x}_t)\|^2] \leq \frac{2\mathbb{E}[F(x_0) - F(x_*)]}{\beta\sqrt{KT}} + \frac{2(3C_g^2\sigma_f^2 + 3C_f^2\sigma_{g'}^2 + C_g^2 L_f^2\sigma_g^2 + 24\gamma\sigma_g^2)}{\sqrt{KT}}$$

$$+ \frac{48C_f^2\sigma_{g'}^2 + 16C_g^2\sigma_f^2 + 99C_g^2 L_f^2\sigma_g^2}{16\beta C_g^2 L_f^2\sqrt{KT}} + \frac{6C_g^2 L_f^2\sigma_g^2}{\beta\gamma\sqrt{KT}} + \frac{48\sigma^2}{\beta\sqrt{KT}}.$$

$$(15)$$

**Remark 1.** *For sufficiently large $T$, Corollary 1 indicates that GP-DSCGD can achieve linear speedup with respect to the number of devices, which is consistent with decentralized SGD.*

**Remark 2.** *To achieve the $\epsilon$-accuracy solution such that $\frac{1}{T}\sum_{t=0}^{T-1}\mathbb{E}[\|\nabla F(\bar{x}_t)\|^2] \leq \epsilon$, the communication complexity is $O\left(\frac{1}{K\epsilon^2}\right)$ and the sample complexity is $T \times B = O\left(\frac{1}{K\epsilon^3}\right)$. When $K = 1$, the sample complexity of our method is better than $O(\frac{1}{\epsilon^3})$ of traditional SCGD [20].*

Note that when $T$ is sufficiently large, $\frac{1}{2\eta L_F}$ could be greater than 1 so that $\beta$ and $\gamma$ do not affect the order of the convergence rate in Corollary 1. Additionally, our method does not use the acceleration or variance reduction techniques as [21, 25, 24] so that the sample complexity of our method is inferior to those methods.

**Theorem 2.** *Assume Assumptions 1-4 hold, for Algorithm 2, by setting $|\mathcal{A}_t^k| = |\mathcal{B}_t^k| = B$, $\gamma > 0$, $\beta_t = \beta \leq \min\{\frac{1}{8\gamma}, \frac{1}{2\eta L_F}, 1\}$, and $\eta \leq \min\{\eta_1, \eta_2, \eta_3\}$, where*

$$\eta_1 = \frac{4\gamma(1-\lambda^2)}{8\gamma^2 L_F^2 + 289C_g^4 L_f^2 + 16C_f^2 L_g^2}, \eta_2 = \frac{\gamma}{2\sqrt{19C_g^4 L_f^2 + C_f^2 L_g^2}},$$

$$\eta_3 = \frac{\sqrt{b^2+4ac}-b}{2a}, a = \frac{27C_g^4 L_f^2}{4\gamma^2} + \frac{3C_f^2 L_g^2}{8\gamma^2} + \frac{3C_g^4 L_f^2}{128\gamma^2}, b = \frac{6}{1-\lambda^2}, c = (1-\lambda)^2,$$

$$(16)$$

*we can get*

$$\frac{1}{T}\sum_{t=0}^{T-1}\mathbb{E}[\|\nabla F(\bar{x}_t)\|^2] \leq \frac{2\mathbb{E}[F(x_0) - F(x_*)]}{\eta\beta T} + \frac{6C_g^2\sigma_f^2 + 6C_f^2\sigma_{g'}^2 + 3C_g^2L_f^2\sigma_g^2}{B}$$
$$+ \frac{24\eta^2(C_f^2\sigma_{g'}^2 + C_g^2\sigma_f^2)(19C_g^4L_f^2 + C_f^2L_g^2)}{\gamma^2 B} + \frac{48\eta^2(C_f^2\sigma_{g'}^2 + C_g^2\sigma_f^2)(19C_g^4L_f^2 + C_f^2L_g^2)}{\gamma^2(1-\lambda^2)B}$$
$$+ \frac{48\eta(C_f^2\sigma_{g'}^2 + C_g^2\sigma_f^2)}{\gamma(1-\lambda^2)B} + \frac{96\eta(C_f^2\sigma_{g'}^2 + C_g^2\sigma_f^2)}{\gamma(1-\lambda^2)^2 B} + \frac{8C_g^2L_f^2\sigma_g^2 + 96(1-\lambda)C_g^2L_f^2\sigma^2}{\gamma\beta B} \;.$$

$$(17)$$

**Corollary 2.** *Assume Assumptions 1-4 hold, for Algorithm 2, by setting $\eta = O(\frac{\sqrt{K}}{\sqrt{T}})$, $B = O(\sqrt{KT})$, $\beta$ and $\gamma$ to be two positive constant values, we can get*

$$\frac{1}{T}\sum_{t=0}^{T-1}\mathbb{E}[\|\nabla F(\bar{x}_t)\|^2] \leq \frac{2\mathbb{E}[F(x_0) - F(x_*)]}{\beta\sqrt{KT}} + \frac{48(C_f^2\sigma_{g'}^2 + C_g^2\sigma_f^2)}{\gamma(1-\lambda^2)T} + \frac{96(C_f^2\sigma_{g'}^2 + C_g^2\sigma_f^2)}{\gamma(1-\lambda^2)^2 T}$$
$$+ \frac{24(C_f^2\sigma_{g'}^2 + C_g^2\sigma_f^2)(19C_g^4L_f^2 + C_f^2L_g^2)}{\gamma^2 K^{1/2}T^{3/2}} + \frac{48(C_f^2\sigma_{g'}^2 + C_g^2\sigma_f^2)(19C_g^4L_f^2 + C_f^2L_g^2)}{\gamma^2(1-\lambda^2)K^{1/2}T^{3/2}}$$
$$+ \frac{6C_g^2\sigma_f^2 + 6C_f^2\sigma_{g'}^2 + 3C_g^2L_f^2\sigma_g^2}{\sqrt{KT}} + \frac{8C_g^2L_f^2\sigma_g^2 + 96(1-\lambda)C_g^2L_f^2\sigma^2}{\gamma\beta\sqrt{KT}} \;.$$

$$(18)$$

**Remark 3.** *For sufficiently large T, Corollary 2 also indicates that GT-DSCGD can achieve linear speedup with respect to the number of devices, which is consistent with the decentralized SGD method. Similar to GP-DSCGD, to achieve the $\epsilon$-accuracy solution such that $\frac{1}{T}\sum_{t=0}^{T-1}\mathbb{E}[\|\nabla F(\bar{x}_t)\|^2] \leq \epsilon$, the communication complexity of GT-DSCGD is $O\left(\frac{1}{K\epsilon^2}\right)$ and the sample complexity is $O\left(\frac{1}{K\epsilon^3}\right)$.*

### 4.3 Proof Sketch

In this subsection, we present the main idea of establishing the convergence rate of Algorithm 1 and Algorithm 2. More details are deferred to Appendix.

**Proof sketch of Theorem 1** To establish the convergence rate of Algorithm 1, we proposed a novel potential function as follows:

$$P_t = \mathbb{E}[F(\bar{x}_t)] + \frac{3\eta C_g^2L_f^2}{\gamma}\frac{1}{K}\sum_{k=1}^{K}\mathbb{E}[\|u_t^{(k)} - g^{(k)}(x_t^{(k)})\|^2] + \frac{4\eta}{K}\sum_{k=1}^{K}\mathbb{E}[\|u_t^{(k)} - \bar{u}_t\|^2]$$
$$+ \frac{\gamma\eta\beta}{4C_g^2L_f^2}\frac{1}{K}\sum_{k=1}^{K}\mathbb{E}[\|z_t^{(k)} - \bar{z}_t\|^2] + \frac{1}{K}\sum_{k=1}^{K}\mathbb{E}[\|x_t^{(k)} - \bar{x}_t\|^2] \;.$$

$$(19)$$

Then, the task boils down to bound each term in this potential function. In particular, we obtain these bounds in Lemmas 3, 4, 5, 6, respectively in Appendix. With these lemmas, by setting the hyperparameter as shown in Theorem 1, we can get

$$P_{t+1} - P_t \leq -\frac{\eta\beta}{2}\mathbb{E}[\|\nabla F(\bar{x}_t)\|^2] + \frac{\eta\beta(3C_g^2\sigma_f^2 + 3C_f^2\sigma_{g'}^2 + C_g^2L_f^2\sigma_g^2 + 24\gamma\sigma_g^2)}{B}$$
$$+ \frac{24C_f^2\sigma_{g'}^2 + 8C_g^2\sigma_f^2 + 3C_g^2L_f^2\sigma_g^2}{B}\frac{\gamma\eta\beta}{4C_g^2L_f^2} \;.$$

$$(20)$$

By summing $t$, we can complete the proof.

**Proof sketch of Theorem 2** Similar to the convergence analysis of Algorithm 1, we also proposed a novel potential function to facilitate the theoretical analysis. Here, different from the proof of Theorem 1, we didn't directly use $\mathbb{E}[\|s_t^{(k)} - \bar{s}_t\|^2]$ in the potential function. Instead, we introduce an additional variable $h_t^{(k)}$, which is defined as follows:

$$h_{-1}^{(k)} = 0 \,, h_t^{(k)} = \sum_{j\in\mathcal{N}_{v_k}} w_{kj}h_{t-1}^{(j)} + q_t^{(k)} - q_{t-1}^{(k)} \,,$$

$$(21)$$

where $q_t^{(k)} = \nabla g^{(k)}(x_t^{(k)})^T \nabla_g f^{(k)}(u_t^{(k)})$ and $q_{-1}^{(k)} = 0$. Here, the difference between $h_t^{(k)}$ and $s_t^{(k)}$ is that $h_t^{(k)}$ uses the full gradient of each function while $s_t^{(k)}$ uses the stochastic gradient. Note that $h_t^{(k)}$ is just used for theoretical analysis. It does NOT need to be computed in practice. Based on this new variable, we define the following potential function:

$$
P_t = \mathbb{E}[F(\bar{x}_t)] + \frac{4\eta C_g^2 L_f^2}{\gamma} \frac{1}{K} \sum_{k=1}^K \mathbb{E}[\|u_t^{(k)} - g^{(k)}(x_t^{(k)})\|^2] + \frac{\eta(1-\lambda)}{\gamma} \frac{1}{K} \sum_{k=1}^K \mathbb{E}[\|h_t^{(k)} - \bar{h}_t\|^2]
$$

$$
+ \frac{1}{\gamma} \frac{1}{K} \sum_{k=1}^K \mathbb{E}[\|x_t^{(k)} - \bar{x}_t\|^2] .
$$

(22)

Similar to the proof of Theorem 1, the task boils down to bound each term in the potential function, respectively. The detailed bounds can be found in Lemmas 9, 13, 11. Then, we can also get the bound for $P_{t+1} - P_t$ and complete the proof by summing $t$ over all iterations like the proof of Theorem 1.

## 5 Experiment

### 5.1 Model-Agnostic Meta-Learning

Model-Agnostic Meta-Learning (MAML) [6] is to learn a meta-initialization model parameter for a set of new tasks. Specifically, it is to optimize the following model:

$$
\min_{x \in \mathbb{R}^d} \mathbb{E}_{i \sim p, D_{\text{test}}^i} l_i \left( \mathbb{E}_{D_{\text{train}}^i} \left( x - \alpha \nabla l_i \left( x, D_{\text{train}}^i \right) \right), D_{\text{test}}^i \right) ,
$$

(23)

where $p$ is the task distribution, $\alpha$ is the learning rate, $l_i$ is the loss function for the $i$-th task, $D_{\text{train}}^i$ is the training set of the $i$-th task and $D_{\text{test}}^i$ is the testing set of the $i$-th task. Note that we omit the superscript $k$ for simplication. MAML can be reformulated as a stochastic compositional optimization problem as follows:

$$
f(y; \zeta) = l_i \left( y, D_{\text{test}}^i \right), \text{ where } \zeta = \left( i, D_{\text{test}}^i \right)
$$
$$
g(x; \xi) = x - \alpha \nabla l_i \left( x, D_{\text{train}}^i \right), \text{ where } \xi = D_{\text{train}}^i .
$$

(24)

Then, we can use the stochastic compostional gradient to optimize MAML. In particular, we can use $u_t$ to track $g(x_t)$ as follows:

$$
u_t = (1 - \gamma\eta_{t-1})u_{t-1} + \gamma\eta_{t-1}g(x_t; B_{\xi,t}) = (1 - \gamma\eta_{t-1})u_{t-1} + \gamma\eta_{t-1}(x_{t-1} - \alpha\nabla l_i \left( x_{t-1}, D_{\text{train}}^i \right)).
$$

(25)

In our experiment, we will use GP-DSCGD and GT-DSCGD to solve the distributed MAML problem, where tasks are distributed on different devices.

### 5.2 Regression Task

Following [6], we verify the performance of our methods with the regression problem. In particular, each task in MAML is a regression task, which maps the input to a sine wave. Different sine waves have different amplitudes and phases. Specifically, to generate a set of sine waves, the amplitude is uniformly drawn from $[0.1, 5.0]$ and the phase is uniformly drawn from $[0, \pi]$. Additionally, the input $a$ is randomly drawn from $[-5.0, 5.0]$. Then, we use a parameterized neural network $f(x; a)$ to approximate the sine wave. In our experiment, $f(x; a)$ has two hidden layers and the dimensionality of different layers is $[1, 40, 40, 1]$. In each hidden layer, we use the ReLU function as the activation function. There are numerous regression tasks, the goal of MAML is to learn a good initialization for the model parameter $x$ such that it can be quickly adapted to new regression tasks and have good performance.

In our experiment, we use four GPUs where each GPU is viewed as a device. These devices compose a ring graph. On each device, the number of tasks in each meta-batch is set to 200. The number of samples in each task for training is set to 10. As for the testing set, the number of tasks is set to 500 and the number of samples in each task is also set to 10. In addition, the number of iterations for adaptation in the training phase is set to 1 while it is set to 10 in the testing phase. To verify the performance of our methods, we compare them with the gossip-based decentralized SGD (DSGD)

method. Specifically, the decentralized SGD method directly uses $u_t = g(x_t)$. In other words, DSGD directly uses $x_{t-1} - \alpha \nabla l_i \left( x_{t-1}, D^i_{\text{train}} \right)$ for adaptation rather than that in Eq. (25). In our experiment, $\alpha$ is set to 0.01. For adaptation, the learning rate $\eta$ of DSGD is set to $\eta = 0.001$. Sine Adam has better performance in MAML [6], we actually use the decentralized Adam in our experiment as the baseline method. Accordingly, for fair comparison, we also use the adaptive learning rate for our two methods based on the stochastic compositional gradient $z_t^{(k)}$ and $s_t^{(k)}$. In addition, we set $\gamma = 3.0, \eta = 0.03, \beta = 0.33$. The reason for this setting is that $x_{t+1}^{(k)} \approx x_t^{(k)} - \beta \eta z_t^{(k)}$. Therefore, the effective learning rate $\beta \eta$ of our methods is approximately equal to that of the baseline method, which is a fair comparison.

In Figure 1(a), we show the average and standard deviation of the training loss across different devices. It can be seen that our two methods converge much faster than DSGD. The reason is that our methods use $u_t$ to track the $g(x_t)$ so that the estimation variance for $g(x_t)$ is smaller than DSGD. Meanwhile, it can be seen that our two methods achieve similar performance. Moreover, in Figure 1(b), we plot the loss function value when using the learned model parameter as the initialization for new tasks and then conducting 10 gradient descent steps. It can be seen that our two methods can achieve better performance in adaptation to the new task.

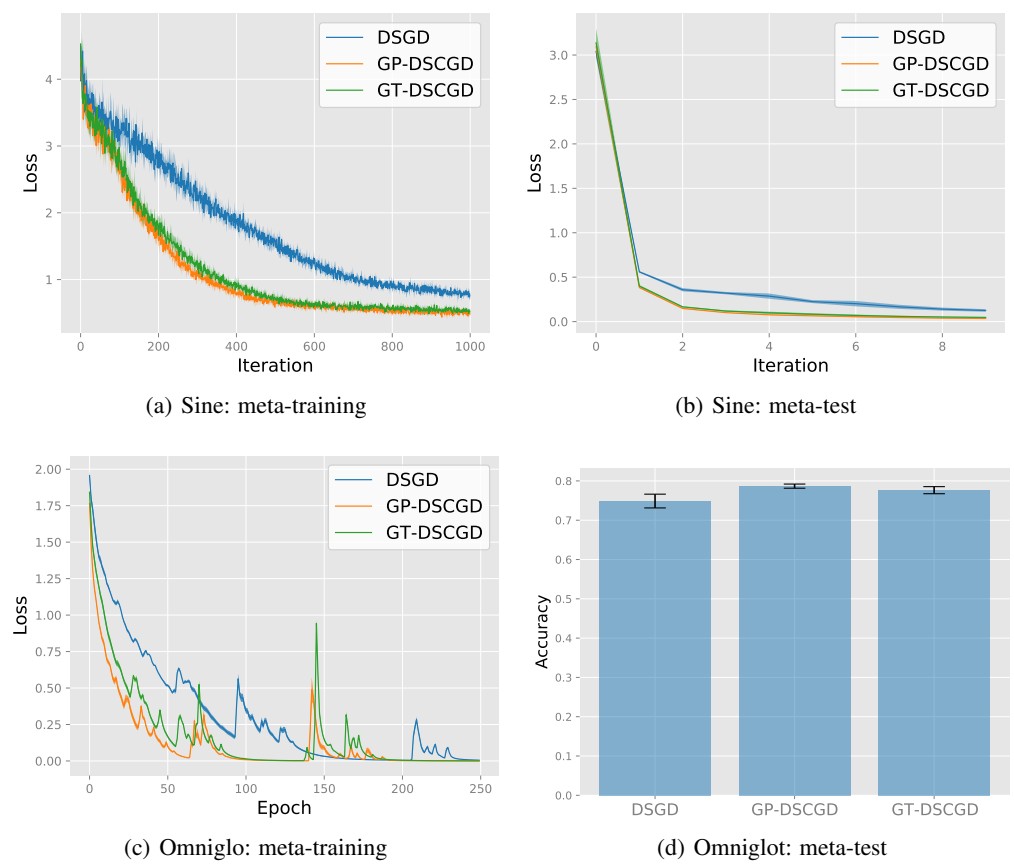

Figure 1: Meta-training and meta-test for the regression and classification task.

## 5.3 Classification Task

Besides the regression task, we also verify the performance of our distributed training methods with the classification task. Here, we use the Omniglot image dataset [12]. It has 1623 characters and each character has 20 images. Under the MAML setting, recognizing each character corresponds to a task so that there are 1623 tasks totally. In our experiment, we use 1200 tasks as the training set and the rest tasks as the testing set. To recognize the character, we use the same convolutional neural network as [6]. To get the meta-initialization for the model parameter, we use the 5-way 1-shot setting. As

for the distributed training, we still use four devices. The meta batch-size on each device is 8. The number of iterations for adaptation in the training phase is set to 1 while it is set to 3 in the testing phase. Additionally, we set the inner learning rate $\alpha = 0.4$, the learning rate of DSGD $\eta = 0.001$. As for our methods, we keep the same learning rates as the regression task.

From Figure 1(c), we also plot the loss function value for the classification task. We can still find that our two methods converge much faster than DSGD, which further confirms the effectiveness of our proposed methods. Additionally, we show the classification accuracy after adaptation for new tasks in Figure 1(d). It is obvious that our two methods can achieve higher accuracy and smaller variance compared with DSGD, which further confirms the effectiveness of our methods.

## 5.4 Additional Experiment

In this experiment, we verify the performance of our methods with different number of devices. In particular, we compare the performance using four GPUs and that using eight GPUs for the regression task. For these two settings, we use the same overall meta-batch size as that in Section 5.2, and then uniformly distribute the tasks on all devices. As for other hyperparameters, they are the same for these two settings. In Figure 2, we plot the training loss function value with respect to the consumed time. It can be observed that our two methods converge much faster when using more devices, which confirms the speedup of the decentralized training.

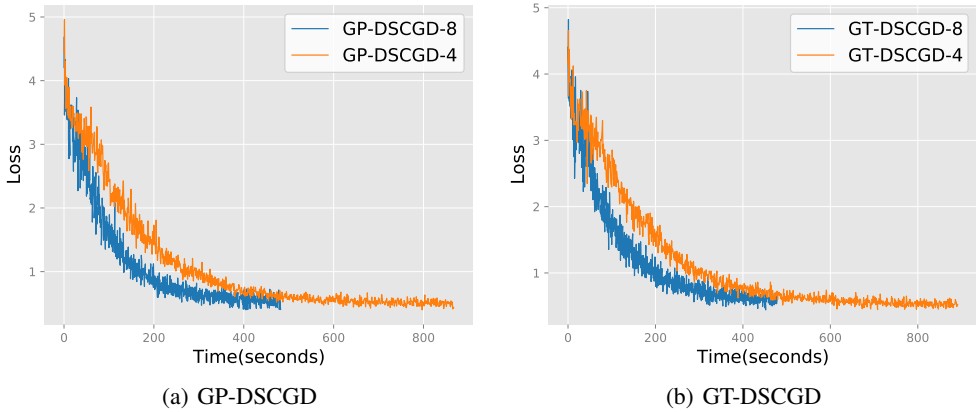

(a) GP-DSCGD          (b) GT-DSCGD

Figure 2: Meta-training for the regression task by using GP-DSCGD and GT-DSCGD.

## 6 Conclusion

In this paper, we studied the distributed training methods for the stochastic compositional optimization problem. In particular, we proposed two novel decentralized stochastic compositional gradient descent method based on the gossip communication mechanism and the gradient tracking communication mechanism. As far as we know, our work is the first one facilitating distributed training for large-scale stochastic compositional optimization problem. Meanwhile, we established the convergence rate of our proposed methods with novel theoretical analysis. The theoretical results indicate that our two methods can achieve linear speedup with respect to the number of devices. This is the first work disclosing this favorable result. At last, extensive experimental results on the MAML task demonstrate the superior performance of our methods.

## Acknowledgments and Disclosure of Funding

This work was partially supported by NSF IIS 1845666, 1852606, 1838627, 1837956, 1956002, OIA 2040588. In addition, this research includes calculations carried out on HPC resources supported in part by the National Science Foundation through major research instrumentation grant number 1625061 and by the US Army Research Laboratory under contract number W911NF-16-2-0189.

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
