# A  Appendix

Throughout the convergence analysis in this paper, we use $\bar{a}_t = \frac{1}{K}\sum_{k=1}^{K} a_t^{(k)}$ to denote the average variables across different devices and introduces two auxiliary matrices $A_t = [a_t^{(1)}, a_t^{(2)}, \cdots, a_t^{(K)}]$ and $\bar{A}_t = [\bar{a}_t, \bar{a}_t, \cdots, \bar{a}_t]$, where $a_t^{(k)} \in \{x_t^{(k)}, \tilde{x}_t^{(k)}, z_t^{(k)}, v_t^{(k)}, u_t^{(k)}, s_t^{(k)}, h_t^{(k)}, q_t^{(k)}\}$.

## A.1  Convergence Analysis of Algorithm 1

**Lemma 1.** *In terms of Assumptions 2-4, by setting $\beta_t \leq \frac{1}{2\eta L_F}$, the following inequality holds.*

$$
\begin{aligned}
\mathbb{E}[F(\bar{x}_{t+1})] \leq{}& \mathbb{E}[F(\bar{x}_t)] - \frac{\beta_t}{2}\mathbb{E}[\|\nabla F(\bar{x}_t)\|^2] - \frac{\eta\beta_t}{4}\mathbb{E}[\|\bar{z}_t\|^2] + \frac{\beta_t L_F^2}{K}\sum_{k=1}^{K}\mathbb{E}[\|x_t^{(k)} - \bar{x}_t\|^2] \\
&+ \frac{3}{K}\sum_{k=1}^{K}\frac{\beta_t C_g^2 \sigma_f^2}{|\mathcal{A}_t^{(k)}|} + \frac{3\beta_t C_g^2 L_f^2}{K}\sum_{k=1}^{K}\mathbb{E}[\|u_t^{(k)} - g^{(k)}(x_t^{(k)})\|^2] + \frac{3}{K}\sum_{k=1}^{K}\frac{\beta_t C_f^2 \sigma_{g'}^2}{|\mathcal{B}_t^{(k)}|}.
\end{aligned}
\tag{26}
$$

*Proof.* In terms of the smoothness of $F(x)$, we can get

$$
\begin{aligned}
F(\bar{x}_{t+1}) \leq{}& F(\bar{x}_t) + \langle\nabla F(\bar{x}_t), \bar{x}_{t+1} - \bar{x}_t\rangle + \frac{L_F}{2}\|\bar{x}_{t+1} - \bar{x}_t\|^2 \\
={}& F(\bar{x}_t) + \beta_t\langle\nabla F(\bar{x}_t), \bar{\tilde{x}}_{t+1} - \bar{x}_t\rangle + \frac{\beta_t^2 L_F}{2}\|\bar{\tilde{x}}_{t+1} - \bar{x}_t\|^2 \\
={}& F(\bar{x}_t) - \eta\beta_t\langle\nabla F(\bar{x}_t), \bar{z}_t\rangle + \frac{\eta^2\beta_t^2 L_F}{2}\|\bar{z}_t\|^2 \\
={}& F(\bar{x}_t) - \frac{\eta\beta_t}{2}\|\nabla F(\bar{x}_t)\|^2 - (\frac{\eta\beta_t}{2} - \frac{\beta_t^2\eta^2 L_F}{2})\|\bar{z}_t\|^2 + \frac{\eta\beta_t}{2}\|\bar{z}_t - \nabla F(\bar{x}_t)\|^2 \\
\leq{}& F(\bar{x}_t) - \frac{\eta\beta_t}{2}\|\nabla F(\bar{x}_t)\|^2 - \frac{\eta\beta_t}{4}\|\bar{z}_t\|^2 + \frac{\eta\beta_t}{2}\|\bar{z}_t - \nabla F(\bar{x}_t)\|^2 \\
\leq{}& F(\bar{x}_t) - \frac{\eta\beta_t}{2}\|\nabla F(\bar{x}_t)\|^2 - \frac{\eta\beta_t}{4}\|\bar{z}_t\|^2 + \eta\beta_t\|\bar{z}_t - \frac{1}{K}\sum_{k=1}^{K}\nabla g^{(k)}(x_t^{(k)})^T\nabla f^{(k)}(g^{(k)}(x_t^{(k)}))\|^2 \\
&+ \eta\beta_t\|\frac{1}{K}\sum_{k=1}^{K}\nabla g^{(k)}(x_t^{(k)})^T\nabla f^{(k)}(g^{(k)}(x_t^{(k)})) - \frac{1}{K}\sum_{k=1}^{K}\nabla g^{(k)}(\bar{x}_t)^T\nabla f^{(k)}(g^{(k)}(\bar{x}_t))\|^2 \\
\leq{}& F(\bar{x}_t) - \frac{\eta\beta_t}{2}\|\nabla F(\bar{x}_t)\|^2 - \frac{\eta\beta_t}{4}\|\bar{z}_t\|^2 + \eta\beta_t\|\bar{z}_t - \frac{1}{K}\sum_{k=1}^{K}\nabla g^{(k)}(x_t^{(k)})^T\nabla f^{(k)}(g^{(k)}(x_t^{(k)}))\|^2 \\
&+ \frac{\eta\beta_t L_F^2}{K}\sum_{k=1}^{K}\|x_t^{(k)} - \bar{x}_t\|^2, \\
\leq{}& F(\bar{x}_t) - \frac{\eta\beta_t}{2}\|\nabla F(\bar{x}_t)\|^2 - \frac{\eta\beta_t}{4}\|\bar{z}_t\|^2 + \frac{\eta\beta_t L_F^2}{K}\sum_{k=1}^{K}\|x_t^{(k)} - \bar{x}_t\|^2 \\
&+ \frac{3}{K}\sum_{k=1}^{K}\frac{\eta\beta_t C_g^2 \sigma_f^2}{|\mathcal{A}_t^{(k)}|} + \frac{3\eta\beta_t C_g^2 L_f^2}{K}\sum_{k=1}^{K}\|u_t^{(k)} - g^{(k)}(x_t^{(k)})\|^2 + \frac{3}{K}\sum_{k=1}^{K}\frac{\eta\beta_t C_f^2 \sigma_{g'}^2}{|\mathcal{B}_t^{(k)}|},
\end{aligned}
\tag{27}
$$

where the first inequality holds due to $\beta_t \leq \frac{1}{2\eta L_F}$, the last inequality holds due to Lemma 2.

$\square$

**Lemma 2.** *In terms of Assumptions 2-4, the following inequality holds.*

$$
\begin{aligned}
&\mathbb{E}[\|\bar{z}_t - \frac{1}{K}\sum_{k=1}^{K}\nabla g^{(k)}(x_t^{(k)})^T\nabla f^{(k)}(g^{(k)}(x_t^{(k)}))\|^2] \\
&\leq \frac{3}{K}\sum_{k=1}^{K}\frac{C_g^2 \sigma_f^2}{|\mathcal{A}_t^{(k)}|} + \frac{3C_g^2 L_f^2}{K}\sum_{k=1}^{K}\mathbb{E}[\|u_t^{(k)} - g^{(k)}(x_t^{(k)})\|^2] + \frac{3}{K}\sum_{k=1}^{K}\frac{C_f^2 \sigma_{g'}^2}{|\mathcal{B}_t^{(k)}|}.
\end{aligned}
\tag{28}
$$

*Proof.* In terms of the definition of $\bar{z}_t$, we can get

$$\mathbb{E}[\|\bar{z}_t - \frac{1}{K}\sum_{k=1}^{K}\nabla g^{(k)}(x_t^{(k)})^T\nabla f^{(k)}(g^{(k)}(x_t^{(k)}))\|^2]$$

$$= \mathbb{E}[\|\frac{1}{K}\sum_{k=1}^{K}(v_t^{(k)})^T\nabla f^{(k)}(u_t^{(k)};\mathcal{A}_t^{(k)}) - \frac{1}{K}\sum_{k=1}^{K}\nabla g^{(k)}(x_t^{(k)})^T\nabla f^{(k)}(g^{(k)}(x_t^{(k)}))\|^2]$$

$$\leq \frac{1}{K}\sum_{k=1}^{K}\mathbb{E}[\|(v_t^{(k)})^T\nabla f^{(k)}(u_t^{(k)};\mathcal{A}_t^{(k)}) - \nabla g^{(k)}(x_t^{(k)})^T\nabla f^{(k)}(g^{(k)}(x_t^{(k)}))\|^2]$$

$$= \frac{1}{K}\sum_{k=1}^{K}\mathbb{E}[\|(v_t^{(k)})^T\nabla f^{(k)}(u_t^{(k)};\mathcal{A}_t^{(k)}) - (v_t^{(k)})^T\nabla f^{(k)}(u_t^{(k)}) + (v_t^{(k)})^T\nabla f^{(k)}(u_t^{(k)})$$
$$- (v_t^{(k)})^T\nabla f^{(k)}(g^{(k)}(x_t^{(k)})) + (v_t^{(k)})^T\nabla f^{(k)}(g^{(k)}(x_t^{(k)})) - \nabla g^{(k)}(x_t^{(k)})^T\nabla f^{(k)}(g^{(k)}(x_t^{(k)}))\|^2]$$

$$\leq \frac{3}{K}\sum_{k=1}^{K}\mathbb{E}[\|v_t^{(k)}\|^2\|\nabla f^{(k)}(u_t^{(k)};\mathcal{A}_t^{(k)}) - \nabla f^{(k)}(u_t^{(k)})\|^2 + \|v_t^{(k)}\|^2\|\nabla f^{(k)}(u_t^{(k)}) - \nabla f^{(k)}(g^{(k)}(x_t^{(k)}))\|^2$$
$$+ \|\nabla f^{(k)}(g^{(k)}(x_t^{(k)}))\|^2\|v_t^{(k)} - \nabla g^{(k)}(x_t^{(k)})\|^2]$$

$$\leq \frac{3}{K}\sum_{k=1}^{K}\frac{C_g^2\sigma_f^2}{|\mathcal{A}_t^{(k)}|} + \frac{3C_g^2L_f^2}{K}\sum_{k=1}^{K}\mathbb{E}[\|u_t^{(k)} - g^{(k)}(x_t^{(k)})\|^2] + \frac{3}{K}\sum_{k=1}^{K}\frac{C_f^2\sigma_{g'}^2}{|\mathcal{B}_t^{(k)}|},$$

$$(29)$$

where the last inequality holds due to Assumptions 2-4. $\square$

**Lemma 3.** *In terms of Assumptions 2-4, by setting $0 < \beta_t \leq \frac{1}{8\gamma}$, for $t > 0$, the following inequality holds.*

$$\mathbb{E}[\|u_t^{(k)} - g^{(k)}(x_t^{(k)})\|^2]$$
$$\leq (1-\gamma\beta_{t-1})\mathbb{E}[\|u_{t-1}^{(k)} - g^{(k)}(x_{t-1}^{(k)})\|^2] + \frac{9\beta_{t-1}C_g^2}{8\gamma}\mathbb{E}[\|\tilde{x}_t^{(k)} - x_{t-1}^{(k)}\|^2] + \frac{\beta_{t-1}^2\gamma^2\sigma_g^2}{|\mathcal{B}_t^{(k)}|}.$$

$$(30)$$

*Proof.* In terms of the definition of $u_t^{(k)}$, for $t > 0$, we can get

$$\mathbb{E}[\|u_t^{(k)} - g^{(k)}(x_t^{(k)})\|^2]$$
$$= \mathbb{E}[\|(1-\gamma\beta_{t-1})u_{t-1}^{(k)} + \gamma\beta_{t-1}g^{(k)}(x_t^{(k)};\mathcal{B}_t^{(k)}) - g^{(k)}(x_t^{(k)})\|^2]$$
$$= \mathbb{E}[\|(1-\gamma\beta_{t-1})(u_{t-1}^{(k)} - g^{(k)}(x_t^{(k)})) + \gamma\beta_{t-1}(g^{(k)}(x_t^{(k)};\mathcal{B}_t^{(k)}) - g^{(k)}(x_t^{(k)}))\|^2] \quad (31)$$
$$\leq (1-\gamma\beta_{t-1})^2\mathbb{E}[\|u_{t-1}^{(k)} - g^{(k)}(x_t^{(k)})\|^2] + \frac{\beta_{t-1}^2\gamma^2\sigma_g^2}{|\mathcal{B}_t^{(k)}|},$$

where the last inequality holds due to Assumption 4. Additionally, we can get

$$(1-\gamma\beta_{t-1})^2\mathbb{E}[\|u_{t-1}^{(k)} - g^{(k)}(x_t^{(k)})\|^2]$$
$$= (1-\gamma\beta_{t-1})^2\mathbb{E}[\|u_{t-1}^{(k)} - g^{(k)}(x_{t-1}^{(k)}) + g^{(k)}(x_{t-1}^{(k)}) - g^{(k)}(x_t^{(k)})\|^2]$$
$$\leq (1-\gamma\beta_{t-1})^2(1+\gamma\beta_{t-1})\mathbb{E}[\|u_{t-1}^{(k)} - g^{(k)}(x_{t-1}^{(k)})\|^2]$$
$$+ (1-\gamma\beta_{t-1})^2(1+\frac{1}{\gamma\beta_{t-1}})\mathbb{E}[\|g^{(k)}(x_{t-1}^{(k)}) - g^{(k)}(x_t^{(k)})\|^2]$$
$$\leq (1-\gamma\beta_{t-1})\mathbb{E}[\|u_{t-1}^{(k)} - g^{(k)}(x_{t-1}^{(k)})\|^2] + \frac{9}{8\gamma\beta_{t-1}}\mathbb{E}[\|g^{(k)}(x_{t-1}^{(k)}) - g^{(k)}(x_t^{(k)})\|^2] \quad (32)$$
$$\leq (1-\gamma\beta_{t-1})\mathbb{E}[\|u_{t-1}^{(k)} - g^{(k)}(x_{t-1}^{(k)})\|^2] + \frac{9C_g^2}{8\gamma\beta_{t-1}}\mathbb{E}[\|x_t^{(k)} - x_{t-1}^{(k)}\|^2]$$
$$= (1-\gamma\beta_{t-1})\mathbb{E}[\|u_{t-1}^{(k)} - g^{(k)}(x_{t-1}^{(k)})\|^2] + \frac{9\beta_{t-1}C_g^2}{8\gamma}\mathbb{E}[\|\tilde{x}_t^{(k)} - x_{t-1}^{(k)}\|^2],$$

where the third step holds due to $0 < \beta_{t-1} \leq \frac{1}{8\gamma}$, the fourth step holds due to Assumption 3. In the last step, $\beta_{t-1}$ (where $\beta_{t-1} < 1$) is moved to the nominator, which could control the variance tightly. That is the reason why we use Line 11 in Algorithm 1. Then, by combining above two inequalities, we complete the proof.

$\square$

**Lemma 4.** *In terms of Assumptions 1-4, the following inequality holds.*

$$\sum_{k=1}^{K} \|u_{t+1}^{(k)} - \bar{u}_{t+1}\|^2 \leq (1-\gamma\beta_t)\sum_{k=1}^{K}\|u_t^{(k)} - \bar{u}_t\|^2 + \frac{6K\gamma\beta_t\sigma_g^2}{|\mathcal{B}_{t+1}^{(k)}|} + 12\gamma\beta_t C_g^2 \sum_{k=1}^{K}\|x_{t+1}^{(k)} - \bar{x}_{t+1}\|^2,$$

$$\sum_{k=1}^{K}\|u_0^{(k)} - \bar{u}_0\|^2 \leq \frac{6K\sigma^2}{|\mathcal{B}_0^{(k)}|}.$$

(33)

*Proof.* In terms of the definition of $u_{t+1}^{(k)}$, we have

$$\sum_{k=1}^{K}\|u_{t+1}^{(k)} - \bar{u}_{t+1}\|^2$$

$$= \|U_{t+1} - \bar{U}_{t+1}\|_F^2$$

$$= \|(1-\gamma\beta_t)U_t + \gamma\beta_t G_{t+1} - (1-\gamma\beta_t)\bar{U}_t - \gamma\beta_t\bar{G}_{t+1}\|_F^2$$

$$\leq (1+a)(1-\gamma\beta_t)^2\|U_t - \bar{U}_t\|_F^2 + (1+\frac{1}{a})\gamma^2\beta_t^2\|G_{t+1} - \bar{G}_{t+1}\|_F^2$$

(34)

$$= (1-\gamma\beta_t)\|U_t - \bar{U}_t\|_F^2 + \gamma\beta_t\|G_{t+1} - \bar{G}_{t+1}\|_F^2$$

$$= (1-\gamma\beta_t)\sum_{k=1}^{K}\|u_t^{(k)} - \bar{u}_t\|^2 + \gamma\beta_t\sum_{k=1}^{K}\|g^{(k)}(x_{t+1}^{(k)};\mathcal{B}_{t+1}^{(k)}) - \frac{1}{K}\sum_{k'=1}^{K}g^{(k')}(x_{t+1}^{(k')};\mathcal{B}_{t+1}^{(k')})\|^2,$$

where $G_t = [g^{(1)}(x_t^{(1)};\mathcal{B}_t^{(1)}), g^{(2)}(x_t^{(2)};\mathcal{B}_t^{(2)}), \cdots, g^{(K)}(x_t^{(K)};\mathcal{B}_t^{(K)})]$, $\bar{G}_t = G_t \mathbf{1}\mathbf{1}^T/K$, the fourth step holds due to $a = \frac{\gamma\beta_t}{1-\gamma\beta_t}$. In the following, we will bound the last term.

$$\sum_{k=1}^{K}\|g^{(k)}(x_{t+1}^{(k)};\mathcal{B}_{t+1}^{(k)}) - \frac{1}{K}\sum_{k'=1}^{K}g^{(k')}(x_{t+1}^{(k')};\mathcal{B}_{t+1}^{(k')})\|^2$$

$$= \sum_{k=1}^{K}\|g^{(k)}(x_{t+1}^{(k)};\mathcal{B}_{t+1}^{(k)}) - g^{(k)}(x_{t+1}^{(k)}) + g^{(k)}(x_{t+1}^{(k)}) - \frac{1}{K}\sum_{k'=1}^{K}g^{(k')}(x_{t+1}^{(k')})$$

$$+ \frac{1}{K}\sum_{k'=1}^{K}g^{(k')}(x_{t+1}^{(k')}) - \frac{1}{K}\sum_{k'=1}^{K}g^{(k')}(x_{t+1}^{(k')};\mathcal{B}_{t+1}^{(k')})\|^2$$

$$\leq 3\sum_{k=1}^{K}\Big(\|g^{(k)}(x_{t+1}^{(k)};\mathcal{B}_{t+1}^{(k)}) - g^{(k)}(x_{t+1}^{(k)})\|^2 + \|g^{(k)}(x_{t+1}^{(k)}) - \frac{1}{K}\sum_{k'=1}^{K}g^{(k')}(x_{t+1}^{(k')})\|^2$$

(35)

$$+ \|\frac{1}{K}\sum_{k'=1}^{K}g^{(k')}(x_{t+1}^{(k')}) - \frac{1}{K}\sum_{k'=1}^{K}g^{(k')}(x_{t+1}^{(k')};\mathcal{B}_{t+1}^{(k')})\|^2\Big)$$

$$\leq 3\sum_{k=1}^{K}\Big(\frac{2\sigma_g^2}{|\mathcal{B}_{t+1}^{(k)}|} + \|g^{(k)}(x_{t+1}^{(k)}) - \frac{1}{K}\sum_{k'=1}^{K}g^{(k')}(x_{t+1}^{(k')})\|^2\Big)$$

$$\leq \frac{6K\sigma_g^2}{|\mathcal{B}_{t+1}^{(k)}|} + 12C_g^2\sum_{k=1}^{K}\|x_{t+1}^{(k)} - \bar{x}_{t+1}\|^2,$$

where the last step holds due to the following inequality.

$$\sum_{k=1}^{K}\|g^{(k)}(x_{t+1}^{(k)}) - \frac{1}{K}\sum_{k'=1}^{K}g^{(k')}(x_{t+1}^{(k')})\|^2$$

$$= \sum_{k=1}^{K}\|g^{(k)}(x_{t+1}^{(k)}) - g^{(k)}(\bar{x}_{t+1}) + g(\bar{x}_{t+1}) - \frac{1}{K}\sum_{k'=1}^{K}g^{(k')}(x_{t+1}^{(k')})\|^2$$

$$\leq 2\sum_{k=1}^{K}\Big(\|g^{(k)}(x_{t+1}^{(k)}) - g^{(k)}(\bar{x}_{t+1})\|^2 + \|\frac{1}{K}\sum_{k'=1}^{K}g^{(k')}(\bar{x}_{t+1}) - \frac{1}{K}\sum_{k'=1}^{K}g^{(k')}(x_{t+1}^{(k')})\|^2\Big)$$

(36)

$$\leq 2\sum_{k=1}^{K}\Big(C_g^2\|x_{t+1}^{(k)} - \bar{x}_{t+1}\|^2 + \frac{1}{K}\sum_{k'=1}^{K}C_g^2\|\bar{x}_{t+1} - x_{t+1}^{(k')}\|^2\Big)$$

$$= 4C_g^2\sum_{k=1}^{K}\|x_{t+1}^{(k)} - \bar{x}_{t+1}\|^2.$$

By combining Eq. (34) and Eq. (35), the proof for the first part is completed.

When $t = 0$, we can get

$$\sum_{k=1}^{K} \|u_0^{(k)} - \bar{u}_0\|^2 = \sum_{k=1}^{K} \|g^{(k)}(x_0^{(k)}; \mathcal{B}_0^{(k)}) - \frac{1}{K} \sum_{k'=1}^{K} g^{(k')}(x_0^{(k')}; \mathcal{B}_0^{(k')})\|^2 \leq \frac{6K\sigma^2}{|\mathcal{B}_0^{(k)}|}, \tag{37}$$

which completes the proof. □

**Lemma 5.** *In terms of Assumptions 2-4, the following inequality holds.*

$$\sum_{k=1}^{K} \|z_{t+1}^{(k)} - \bar{z}_{t+1}\|^2 \leq 48 C_f^2 L_g^2 \sum_{k=1}^{K} \|x_{t+1}^{(k)} - \bar{x}_{t+1}\|^2 + 16 \sum_{k=1}^{K} C_g^2 L_f^2 \|u_{t+1}^{(k)} - \bar{u}_{t+1}\|^2$$
$$+ \frac{24 K C_f^2 \sigma_{g'}^2}{|\mathcal{B}_{t+1}^{(k)}|} + \frac{8 K C_g^2 \sigma_f^2}{|\mathcal{A}_{t+1}^{(k)}|}, \tag{38}$$
$$\sum_{k=1}^{K} \|z_0^{(k)} - \bar{z}_0\|^2 \leq \frac{24 K C_f^2 \sigma_{g'}^2}{|\mathcal{B}_0^{(k)}|} + \frac{8 K C_g^2 \sigma_f^2}{|\mathcal{A}_0^{(k)}|} + \frac{96 K C_g^2 L_f^2 \sigma^2}{|\mathcal{B}_0^{(k)}|}.$$

*Proof.* In terms of the definition of $z_t^{(k)}$, we can get

$$\sum_{k=1}^{K} \|z_{t+1}^{(k)} - \bar{z}_{t+1}\|^2$$
$$= \sum_{k=1}^{K} \|(v_{t+1}^{(k)})^T \nabla f^{(k)}(u_{t+1}^{(k)}; \mathcal{A}_{t+1}^{(k)}) - \frac{1}{K} \sum_{k'=1}^{K} (v_{t+1}^{(k')})^T \nabla f^{(k')}(u_{t+1}^{(k')}; \mathcal{A}_{t+1}^{(k')})\|^2$$
$$= \sum_{k=1}^{K} \|(v_{t+1}^{(k)})^T \nabla f^{(k)}(u_{t+1}^{(k)}; \mathcal{A}_{t+1}^{(k)}) - \frac{1}{K} \sum_{k'=1}^{K} (v_{t+1}^{(k')})^T \nabla f^{(k)}(u_{t+1}^{(k)}; \mathcal{A}_{t+1}^{(k)})$$
$$+ \frac{1}{K} \sum_{k'=1}^{K} (v_{t+1}^{(k')})^T \nabla f^{(k)}(u_{t+1}^{(k)}; \mathcal{A}_{t+1}^{(k)}) - \frac{1}{K} \sum_{k'=1}^{K} (v_{t+1}^{(k')})^T \nabla f^{(k)}(u_{t+1}^{(k)})$$
$$+ \frac{1}{K} \sum_{k'=1}^{K} (v_{t+1}^{(k')})^T \nabla f^{(k)}(u_{t+1}^{(k)}) - \frac{1}{K} \sum_{k'=1}^{K} (v_{t+1}^{(k')})^T \nabla f^{(k')}(u_{t+1}^{(k')}) \tag{39}$$
$$+ \frac{1}{K} \sum_{k'=1}^{K} (v_{t+1}^{(k')})^T \nabla f^{(k')}(u_{t+1}^{(k')}) - \frac{1}{K} \sum_{k'=1}^{K} (v_{t+1}^{(k')})^T \nabla f^{(k')}(u_{t+1}^{(k')}; \mathcal{A}_{t+1}^{(k')})\|^2$$
$$\leq 4 \sum_{k=1}^{K} C_f^2 \|v_{t+1}^{(k)} - \frac{1}{K} \sum_{k'=1}^{K} v_{t+1}^{(k')}\|^2 + \frac{4}{K} \sum_{k=1}^{K} \sum_{k'=1}^{K} C_g^2 \|\nabla f^{(k)}(u_{t+1}^{(k)}; \mathcal{A}_{t+1}^{(k)}) - \nabla f^{(k)}(u_{t+1}^{(k)})\|^2$$
$$+ \frac{4}{K} \sum_{k=1}^{K} \sum_{k'=1}^{K} C_g^2 \|\nabla f^{(k)}(u_{t+1}^{(k)}) - \nabla f^{(k')}(u_{t+1}^{(k')})\|^2$$
$$+ \frac{4}{K} \sum_{k=1}^{K} \sum_{k'=1}^{K} C_g^2 \|\nabla f^{(k')}(u_{t+1}^{(k')}) - \nabla f^{(k')}(u_{t+1}^{(k')}; \mathcal{A}_{t+1}^{(k')})\|^2$$
$$\leq 4 \sum_{k=1}^{K} C_f^2 \|v_{t+1}^{(k)} - \frac{1}{K} \sum_{k'=1}^{K} v_{t+1}^{(k')}\|^2 + \frac{4 K C_g^2 \sigma_f^2}{|\mathcal{A}_{t+1}^{(k)}|} + 16 \sum_{k=1}^{K} C_g^2 L_f^2 \|u_{t+1}^{(k)} - \bar{u}_{t+1}\|^2 + \frac{4 K C_g^2 \sigma_f^2}{|\mathcal{A}_{t+1}^{(k)}|}.$$

Then, we will bound $\sum_{k=1}^{K} \|v_{t+1}^{(k)} - \frac{1}{K}\sum_{k'=1}^{K} v_{t+1}^{(k')}\|^2$ as follows.

$$
\sum_{k=1}^{K} \|v_{t+1}^{(k)} - \frac{1}{K}\sum_{k'=1}^{K} v_{t+1}^{(k')}\|^2
$$

$$
= \sum_{k=1}^{K} \|\nabla g^{(k)}(x_{t+1}^{(k)}; \mathcal{B}_{t+1}^{(k)}) - \frac{1}{K}\sum_{k'=1}^{K} \nabla g^{(k')}(x_{t+1}^{(k')}; \mathcal{B}_{t+1}^{(k')})\|^2
$$

$$
= \sum_{k=1}^{K} \|\nabla g^{(k)}(x_{t+1}^{(k)}; \mathcal{B}_{t+1}^{(k)}) - \nabla g^{(k)}(x_{t+1}^{(k)}) + \nabla g^{(k)}(x_{t+1}^{(k)})
$$

$$
- \frac{1}{K}\sum_{k'=1}^{K} \nabla g^{(k')}(x_{t+1}^{(k')}) + \frac{1}{K}\sum_{k'=1}^{K} \nabla g^{(k')}(x_{t+1}^{(k')}) - \frac{1}{K}\sum_{k'=1}^{K} \nabla g^{(k')}(x_{t+1}^{(k')}; \mathcal{B}_{t+1}^{(k')})\|^2
$$
(40)

$$
\leq 3\sum_{k=1}^{K} \Big( \|\nabla g^{(k)}(x_{t+1}^{(k)}; \mathcal{B}_{t+1}^{(k)}) - \nabla g^{(k)}(x_{t+1}^{(k)})\|^2 + \|\nabla g^{(k)}(x_{t+1}^{(k)}) - \frac{1}{K}\sum_{k'=1}^{K} \nabla g^{(k')}(x_{t+1}^{(k')})\|^2
$$

$$
+ \|\frac{1}{K}\sum_{k'=1}^{K} \nabla g^{(k')}(x_{t+1}^{(k')}) - \frac{1}{K}\sum_{k'=1}^{K} \nabla g^{(k')}(x_{t+1}^{(k')}; \mathcal{B}_{t+1}^{(k')})\|^2 \Big)
$$

$$
\leq 3\sum_{k=1}^{K} \Big( \frac{2\sigma_{g'}^2}{|\mathcal{B}_{t+1}^{(k)}|} + \|\nabla g^{(k)}(x_{t+1}^{(k)}) - \frac{1}{K}\sum_{k'=1}^{K} \nabla g^{(k')}(x_{t+1}^{(k')})\|^2 \Big)
$$

$$
\leq \frac{6K\sigma_{g'}^2}{|\mathcal{B}_{t+1}^{(k)}|} + 12L_g^2 \sum_{k=1}^{K} \|x_{t+1}^{(k)} - \bar{x}_{t+1}\|^2 ,
$$

where the last step holds due to the following inequality.

$$
\sum_{k=1}^{K} \|\nabla g^{(k)}(x_{t+1}^{(k)}) - \frac{1}{K}\sum_{k'=1}^{K} \nabla g^{(k')}(x_{t+1}^{(k')})\|^2
$$

$$
= \sum_{k=1}^{K} \|\nabla g^{(k)}(x_{t+1}^{(k)}) - \nabla g^{(k)}(\bar{x}_{t+1}) + \nabla g(\bar{x}_{t+1}) - \frac{1}{K}\sum_{k'=1}^{K} \nabla g^{(k')}(x_{t+1}^{(k')})\|^2
$$

$$
\leq 2\sum_{k=1}^{K} \Big( \|\nabla g^{(k)}(x_{t+1}^{(k)}) - \nabla g^{(k)}(\bar{x}_{t+1})\|^2 + \|\frac{1}{K}\sum_{k'=1}^{K} \nabla g^{(k')}(\bar{x}_{t+1}) - \frac{1}{K}\sum_{k'=1}^{K} \nabla g^{(k')}(x_{t+1}^{(k')})\|^2 \Big) \quad (41)
$$

$$
\leq 2\sum_{k=1}^{K} \Big( L_g^2 \|x_{t+1}^{(k)} - \bar{x}_{t+1}\|^2 + \frac{1}{K}\sum_{k'=1}^{K} L_g^2 \|\bar{x}_{t+1} - x_{t+1}^{(k')}\|^2 \Big)
$$

$$
= 4L_g^2 \sum_{k=1}^{K} \|x_{t+1}^{(k)} - \bar{x}_{t+1}\|^2 .
$$

By combining Eq. (39) and Eq. (40), the proof for the first part is completed.

When $t = 0$, we can get

$$
\sum_{k=1}^{K} \|z_0^{(k)} - \bar{z}_0\|^2 \leq 4\sum_{k=1}^{K} C_f^2 \|v_0^{(k)} - \frac{1}{K}\sum_{k'=1}^{K} v_0^{(k')}\|^2 + \frac{8KC_g^2\sigma_f^2}{|\mathcal{A}_0^{(k)}|} + 16\sum_{k=1}^{K} C_g^2 L_f^2 \|u_0^{(k)} - \bar{u}_0\|^2
$$
(42)
$$
\leq \frac{24KC_f^2\sigma_{g'}^2}{|\mathcal{B}_0^{(k)}|} + \frac{8KC_g^2\sigma_f^2}{|\mathcal{A}_0^{(k)}|} + \frac{96KC_g^2 L_f^2\sigma^2}{|\mathcal{B}_0^{(k)}|} .
$$

$\square$

**Lemma 6.** *In terms of Assumption 1, the following inequality holds.*

$$
\sum_{k=1}^{K} \|x_{t+1}^{(k)} - \bar{x}_{t+1}\|^2 \leq \Big(1 - \beta_t + \frac{\beta_t(1+\lambda^2)}{2}\Big)\sum_{k=1}^{K} \|x_t^{(k)} - \bar{x}_t\|^2 + \frac{2\beta_t\eta^2}{1-\lambda^2}\sum_{k=1}^{K} \|z_t^{(k)} - \bar{z}_t\|^2 . \quad (43)
$$

*Proof.* In terms of the definition of $x_{t+1}^{(k)}$, we have

$$
\begin{aligned}
& \sum_{k=1}^{K} \|x_{t+1}^{(k)} - \bar{x}_{t+1}\|^2 \\
&= \|X_{t+1} - \bar{X}_{t+1}\|_F^2 \\
&= \|(1 - \beta_t)X_t + \beta_t \tilde{X}_{t+1} - (1 - \beta_t)\bar{X}_t - \beta_t \bar{\tilde{X}}_{t+1}\|_F^2 \\
&\leq (1 + a)(1 - \beta_t)^2 \|X_t - \bar{X}_t\|_F^2 + (1 + \frac{1}{a})\beta_t^2 \|\tilde{X}_{t+1} - \bar{\tilde{X}}_{t+1}\|_F^2 \\
&\leq (1 - \beta_t)\|X_t - \bar{X}_t\|_F^2 + \beta_t \|\tilde{X}_{t+1} - \bar{\tilde{X}}_{t+1}\|_F^2 \\
&= (1 - \beta_t)\|X_t - \bar{X}_t\|_F^2 + \beta_t \|X_t W + \eta Z_t - \bar{X}_t - \eta \bar{Z}_t\|_F^2 \\
&\leq (1 - \beta_t)\|X_t - \bar{X}_t\|_F^2 + (1 + a')\beta_t \|X_t W - \bar{X}_t\|_F^2 + (1 + \frac{1}{a'})\beta_t \eta^2 \|Z_t - \bar{Z}_t\|_F^2 \\
&\leq \left(1 - \beta_t + \frac{\beta_t(1 + \lambda^2)}{2}\right) \sum_{k=1}^{K} \|x_t^{(k)} - \bar{x}_t\|^2 + \frac{2\beta_t \eta^2}{1 - \lambda^2} \sum_{k=1}^{K} \|z_t^{(k)} - \bar{z}_t\|^2 ,
\end{aligned}
\tag{44}
$$

where the second inequality follows from $a = \frac{\beta_t}{1 - \beta_t}$, the last inequality follows from $a' = \frac{1 - \lambda^2}{2\lambda^2}$. $\qquad\square$

**Lemma 7.** *In terms of Assumption 1, the following inequality holds.*

$$
\sum_{k=1}^{K} \|\tilde{x}_{t+1}^{(k)} - x_t^{(k)}\|^2 \leq 8 \sum_{k=1}^{K} \|x_t^{(k)} - \bar{x}_t\|^2 + 4\eta^2 \sum_{k=1}^{K} \|z_t^{(k)} - \bar{z}_t\|^2 + 4\eta^2 K \|\bar{z}_t\|^2
\tag{45}
$$

*Proof.* In terms of the definition of $\tilde{x}_{t+1}^{(k)}$, we can get

$$
\begin{aligned}
& \sum_{k=1}^{K} \|\tilde{x}_{t+1}^{(k)} - x_t^{(k)}\|^2 \\
&= \|\tilde{X}_{t+1} - X_t\|_F^2 \\
&= \|X_t W - \eta Z_t - X_t\|_F^2 \\
&\leq 2\|X_t(W - I)\|_F^2 + 2\eta^2 \|Z_t\|_F^2 \\
&= 2\|(X_t - \bar{X}_t)(W - I)\|_F^2 + 2\eta^2 \|Z_t - \bar{Z}_t + \bar{Z}_t\|_F^2 \\
&\leq 2\|X_t - \bar{X}_t\|_F^2 \|W - I\|_2^2 + 2\eta^2 \|Z_t - \bar{Z}_t + \bar{Z}_t\|_F^2 \\
&\leq 8\|X_t - \bar{X}_t\|_F^2 + 4\eta^2 \|Z_t - \bar{Z}_t\|_F^2 + 4\eta^2 \|\bar{Z}_t\|_F^2 \\
&= 8 \sum_{k=1}^{K} \|x_t^{(k)} - \bar{x}_t\|^2 + 4\eta^2 \sum_{k=1}^{K} \|z_t^{(k)} - \bar{z}_t\|^2 + 4\eta^2 K \|\bar{z}_t\|^2
\end{aligned}
\tag{46}
$$

where the second inequality holds due to $\|AB\|_F \leq \|A\|_2 \|B\|_F$ and the last inequality holds due to $\|I - W\|_2 \leq 2$. $\qquad\square$

To prove Theorem 1, we set $\beta_t = \beta$, $|\mathcal{A}_t^k| = |\mathcal{B}_t^k| = B$, and introduce a potential function, which is defined as follows:

$$
\begin{aligned}
P_t &= \mathbb{E}[F(\bar{x}_t)] + \frac{3\eta C_g^2 L_f^2}{\gamma} \frac{1}{K} \sum_{k=1}^{K} \mathbb{E}[\|u_t^{(k)} - g^{(k)}(x_t^{(k)})\|^2] + \frac{4\eta}{K} \sum_{k=1}^{K} \mathbb{E}[\|u_t^{(k)} - \bar{u}_t\|^2] \\
&\quad + \frac{\gamma \eta \beta}{4C_g^2 L_f^2} \frac{1}{K} \sum_{k=1}^{K} \mathbb{E}[\|z_t^{(k)} - \bar{z}_t\|^2] + \frac{1}{K} \sum_{k=1}^{K} \mathbb{E}[\|x_t^{(k)} - \bar{x}_t\|^2] .
\end{aligned}
\tag{47}
$$

Then, we are ready to prove Theorem 1.

*Proof.* In terms of Lemmas 1, 3, 4, 5, 6, we can get

$$
\begin{aligned}
&P_{t+1} - P_t \\
&\leq -\frac{\eta\beta}{2}\mathbb{E}[\|\nabla F(\bar{x}_t)\|^2] + \Big(\frac{27\eta^3\beta C_g^2 L_f^2 C_g^2}{2\gamma^2} - \frac{\eta\beta}{4}\Big)\mathbb{E}[\|\bar{z}_t\|^2] \\
&\quad + \frac{\eta\beta(3C_g^2\sigma_f^2 + 3C_f^2\sigma_{g'}^2 + 3/8 C_g^2 L_f^2\sigma_g^2 + 24\gamma\sigma_g^2)}{B} + \frac{24C_f^2\sigma_{g'}^2 + 8C_g^2\sigma_f^2 + 3C_g^2 L_f^2\sigma_g^2}{B}\frac{\gamma\eta\beta}{4C_g^2 L_f^2} \\
&\quad + \Big(3\eta\beta C_g^2 L_f^2 - 3\eta\beta C_g^2 L_f^2\Big)\frac{1}{K}\sum_{k=1}^{K}\mathbb{E}[\|u_t^{(k)} - g^{(k)}(x_t^{(k)})\|^2] \\
&\quad + \Big(\frac{27\eta\beta C_g^4 L_f^2}{\gamma^2} + \eta\beta L_F^2 + \frac{\gamma\eta\beta(48 C_f^2 L_g^2 + 48\gamma\beta C_g^4 L_f^2)}{4C_g^2 L_f^2}\Big(1 - \beta + \frac{\beta(1+\lambda^2)}{2}\Big) \\
&\quad + \Big(-\beta + \frac{\beta(1+\lambda^2)}{2}\Big) + 48\gamma\eta\beta C_g^2\Big(1 - \beta + \frac{\beta(1+\lambda^2)}{2}\Big)\Big)\frac{1}{K}\sum_{k=1}^{K}\mathbb{E}[\|x_t^{(k)} - \bar{x}_t\|^2] \\
&\quad + \Big(4\gamma\eta\beta(1 - \gamma\beta) - 4\gamma\eta\beta\Big)\frac{1}{K}\sum_{k=1}^{K}\mathbb{E}[\|u_t^{(k)} - \bar{u}_t\|^2] \\
&\quad + \Big(\frac{\gamma\eta\beta}{4C_g^2 L_f^2}\Big((48 C_f^2 L_g^2 + 48\gamma\beta C_g^4 L_f^2)\frac{2\beta\eta^2}{1-\lambda^2} - 1\Big) + \frac{2\beta\eta^2}{1-\lambda^2} + \frac{96\gamma\eta\beta^2 C_g^2\eta^2}{1-\lambda^2} \\
&\quad + \frac{9\eta^2\beta C_g^2}{2\gamma}\frac{3\eta C_g^2 L_f^2}{\gamma}\Big)\frac{1}{K}\sum_{k=1}^{K}\mathbb{E}[\|z_t^{(k)} - \bar{z}_t\|^2].
\end{aligned}
\tag{48}
$$

By setting $\beta \leq \min\{\frac{1}{8\gamma}, \frac{1}{2\eta L_F}, 1\}, \gamma > 0, \eta \leq \min\{\eta_1, \eta_2, \eta_3\}$ where

$$
\begin{aligned}
\eta_1 &= \frac{\gamma}{3\sqrt{6}C_g^2 L_f}, \\
\eta_2 &= \frac{1-\lambda^2}{2}\Big/\Big(\frac{27 C_g^4 L_f^2 + \gamma^2 L_F^2}{\gamma^2} + \frac{\gamma(24 C_f^2 L_g^2 + 99 C_g^4 L_f^2)}{2C_g^2 L_f^2}\Big) \\
\eta_3 &= \frac{\sqrt{b^2 + 4ac} - b}{2a}, a = \frac{(24 C_f^2 L_g^2 + 3 C_g^4 L_f^2)}{8(1-\lambda^2)C_g^2 L_f^2} + \frac{12 C_g^2}{1-\lambda^2} + \frac{27 C_g^4 L_f^2}{2\gamma^2}, b = \frac{2}{1-\lambda^2}, c = \frac{\gamma}{4C_g^2 L_f^2},
\end{aligned}
\tag{49}
$$

we can get

$$
\begin{aligned}
P_{t+1} - P_t &\leq -\frac{\eta\beta}{2}\mathbb{E}[\|\nabla F(\bar{x}_t)\|^2] \\
&\quad + \frac{\eta\beta(3C_g^2\sigma_f^2 + 3C_f^2\sigma_{g'}^2 + 3/8 C_g^2 L_f^2\sigma_g^2 + 24\gamma\sigma_g^2)}{B} + \frac{24C_f^2\sigma_{g'}^2 + 8C_g^2\sigma_f^2 + 3C_g^2 L_f^2\sigma_g^2}{B}\frac{\gamma\eta\beta}{4C_g^2 L_f^2}.
\end{aligned}
\tag{50}
$$

By summing over $t$ from 0 to $T - 1$, we can get

$$
\begin{aligned}
&\frac{1}{T}\sum_{t=0}^{T-1}\frac{\eta\beta}{2}\mathbb{E}[\|\nabla F(\bar{x}_t)\|^2] \\
&\leq \frac{P_0 - P_T}{T} + \frac{\eta\beta(3C_g^2\sigma_f^2 + 3C_f^2\sigma_{g'}^2 + C_g^2 L_f^2\sigma_g^2 + 24\gamma\sigma_g^2)}{B} + \frac{24C_f^2\sigma_{g'}^2 + 8C_g^2\sigma_f^2 + 3C_g^2 L_f^2\sigma_g^2}{B}\frac{\gamma\eta\beta}{4C_g^2 L_f^2} \\
&\leq \frac{\mathbb{E}[F(\bar{x}_0) - F(x_*)]}{T} + \frac{\eta\beta(3C_g^2\sigma_f^2 + 3C_f^2\sigma_{g'}^2 + C_g^2 L_f^2\sigma_g^2 + 24\gamma\sigma_g^2)}{B} + \frac{24C_f^2\sigma_{g'}^2 + 8C_g^2\sigma_f^2 + 3C_g^2 L_f^2\sigma_g^2}{B}\frac{\gamma\eta\beta}{4C_g^2 L_f^2} \\
&\quad + \frac{3\eta C_g^2 L_f^2\sigma_g^2}{\gamma B} + \frac{24\eta\sigma^2}{B} + \frac{\gamma\eta\beta}{C_g^2 L_f^2}\frac{6C_f^2\sigma_{g'}^2 + 2C_g^2\sigma_f^2 + 24 C_g^2 L_f^2\sigma^2}{B},
\end{aligned}
\tag{51}
$$

where the last step holds due to $P_T \geq F(\bar{x}_T) \geq F(x_*)$ and

$$P_0 = \mathbb{E}[F(\bar{x}_0)] + \frac{3\eta C_g^2 L_f^2}{\gamma} \frac{1}{K} \sum_{k=1}^{K} \mathbb{E}[\|u_0^{(k)} - g^{(k)}(x_0^{(k)})\|^2] + \frac{4\eta}{K} \sum_{k=1}^{K} \mathbb{E}[\|u_0^{(k)} - \bar{u}_0\|^2]$$

$$+ \frac{\gamma\eta\beta}{4C_g^2 L_f^2} \frac{1}{K} \sum_{k=1}^{K} \mathbb{E}[\|z_0^{(k)} - \bar{z}_0\|^2] + \frac{1}{K} \sum_{k=1}^{K} \mathbb{E}[\|x_0^{(k)} - \bar{x}_0\|^2] \tag{52}$$

$$\leq \mathbb{E}[F(\bar{x}_0)] + \frac{3\eta C_g^2 L_f^2 \sigma_g^2}{\gamma B} + \frac{24\eta\sigma^2}{B} + \frac{\gamma\eta\beta}{C_g^2 L_f^2} \frac{6C_f^2 \sigma_{g'}^2 + 2C_g^2 \sigma_f^2 + 24C_g^2 L_f^2 \sigma^2}{B} .$$

By dividing $\frac{\eta\beta}{2}$ on both sides of the previous inequality, we can get

$$\frac{1}{T} \sum_{t=0}^{T-1} \mathbb{E}[\|\nabla F(\bar{x}_t)\|^2]$$

$$\leq \frac{2\mathbb{E}[F(\bar{x}_0) - F(x_*)]}{\eta\beta T} + \frac{2(3C_g^2 \sigma_f^2 + 3C_f^2 \sigma_{g'}^2 + C_g^2 L_f^2 \sigma_g^2 + 24\gamma\sigma_g^2)}{B} \tag{53}$$

$$+ \frac{48C_f^2 \sigma_{g'}^2 + 16C_g^2 \sigma_f^2 + 99C_g^2 L_f^2 \sigma_g^2}{16\beta C_g^2 L_f^2 B} + \frac{6C_g^2 L_f^2 \sigma_g^2}{\beta\gamma B} + \frac{48\sigma^2}{\beta B} .$$

$\square$

## A.2 Convergence Analysis of Algorithm 2

**Lemma 8.** *In terms of Assumptions 2-4, by setting $\beta_t \leq \frac{1}{2\eta L_F}$, the following inequality holds.*

$$\mathbb{E}[F(\bar{x}_{t+1})] \leq \mathbb{E}[F(\bar{x}_t)] - \frac{\eta\beta_t}{2} \mathbb{E}[\|\nabla F(\bar{x}_t)\|^2] - \frac{\eta\beta_t}{4} \mathbb{E}[\|\bar{s}_t\|^2] + \frac{\eta\beta_t L_F^2}{K} \sum_{k=1}^{K} \mathbb{E}[\|x_t^{(k)} - \bar{x}_t\|^2]$$

$$+ \frac{3}{K} \sum_{k=1}^{K} \frac{\eta\beta_t C_g^2 \sigma_f^2}{|\mathcal{A}_t^{(k)}|} + \frac{3\eta\beta_t C_g^2 L_f^2}{K} \sum_{k=1}^{K} \mathbb{E}[\|u_t^{(k)} - g^{(k)}(x_t^{(k)})\|^2] + \frac{3}{K} \sum_{k=1}^{K} \frac{\eta\beta_t C_f^2 \sigma_{g'}^2}{|\mathcal{B}_t^{(k)}|} . \tag{54}$$

**Lemma 9.** *In terms of Assumptions 2-4, by setting $0 < \beta_t \leq \frac{1}{8\gamma}$, the following inequality holds.*

$$\mathbb{E}[\|u_{t+1}^{(k)} - g^{(k)}(x_{t+1}^{(k)})\|^2]$$

$$\leq (1 - \gamma\beta_t)\mathbb{E}[\|u_t^{(k)} - g^{(k)}(x_t^{(k)})\|^2] + \frac{9\beta_t C_g^2}{8\gamma} \mathbb{E}[\|\tilde{x}_{t+1}^{(k)} - x_t^{(k)}\|^2] + \frac{\beta_t^2 \gamma^2 \sigma_g^2}{|\mathcal{B}_{t+1}^{(k)}|} . \tag{55}$$

**Lemma 10.** *In terms of Assumption 1, the following inequality holds.*

$$\sum_{k=1}^{K} \mathbb{E}[\|\tilde{x}_{t+1}^{(k)} - x_t^{(k)}\|^2] \leq 8 \sum_{k=1}^{K} \mathbb{E}[\|x_t^{(k)} - \bar{x}_t\|^2] + 4\eta^2 \sum_{k=1}^{K} \mathbb{E}[\|s_t^{(k)} - \bar{s}_t\|^2] + 4\eta^2 K \mathbb{E}[\|\bar{s}_t\|^2] . \tag{56}$$

**Lemma 11.** *In terms of Assumption 1, the following inequality holds.*

$$\sum_{k=1}^{K} \mathbb{E}[\|x_{t+1}^{(k)} - \bar{x}_{t+1}\|^2] \leq \left(1 - \beta_t + \frac{\beta_t(1 + \lambda^2)}{2}\right) \sum_{k=1}^{K} \mathbb{E}[\|x_t^{(k)} - \bar{x}_t\|^2] + \frac{2\beta_t \eta^2}{1 - \lambda^2} \sum_{k=1}^{K} \mathbb{E}[\|s_t^{(k)} - \bar{s}_t\|^2] . \tag{57}$$

The above four lemmas can be proved by exactly following Lemmas 1, 3, 7, 6, respectively. Thus, we omit their proof.

**Lemma 12.** *In terms of Assumption 2-4, the following inequality holds.*

$$\sum_{k=1}^{K} \mathbb{E}[\|u_{t+1}^{(k)} - u_t^{(k)}\|^2] \leq \sum_{k=1}^{K} 3\gamma^2 \beta_t^2 \left(\mathbb{E}[\|u_t^{(k)} - g^{(k)}(x_t^{(k)})\|^2] + C_g^2 \beta_t^2 \mathbb{E}[\|\tilde{x}_{t+1}^{(k)} - x_t^{(k)}\|^2] + \frac{\sigma_g^2}{|\mathcal{B}_{t+1}^{(k)}|}\right) . \tag{58}$$

*Proof.* In terms of the definition of $u_t^{(k)}$, we can get

$$\sum_{k=1}^{K} \mathbb{E}[\|u_{t+1}^{(k)} - u_t^{(k)}\|^2]$$

$$= \sum_{k=1}^{K} \gamma^2 \beta_t^2 \mathbb{E}[\|u_t^{(k)} - g^{(k)}(x_{t+1}^{(k)}; \mathcal{B}_{t+1}^{(k)})\|^2]$$

$$= \sum_{k=1}^{K} \gamma^2 \beta_t^2 \mathbb{E}[\|u_t^{(k)} - g^{(k)}(x_t^{(k)}) + g^{(k)}(x_t^{(k)}) - g^{(k)}(x_{t+1}^{(k)}) + g^{(k)}(x_{t+1}^{(k)}) - g^{(k)}(x_{t+1}^{(k)}; \mathcal{B}_{t+1}^{(k)})\|^2] \tag{59}$$

$$\leq \sum_{k=1}^{K} 3\gamma^2 \beta_t^2 \Big( \mathbb{E}[\|u_t^{(k)} - g^{(k)}(x_t^{(k)})\|^2] + \mathbb{E}[\|g^{(k)}(x_t^{(k)}) - g^{(k)}(x_{t+1}^{(k)})\|^2]$$

$$+ \mathbb{E}[\|g^{(k)}(x_{t+1}^{(k)}) - g^{(k)}(x_{t+1}^{(k)}; \mathcal{B}_{t+1}^{(k)})\|^2] \Big)$$

$$\leq \sum_{k=1}^{K} 3\gamma^2 \beta_t^2 \Big( \mathbb{E}[\|u_t^{(k)} - g^{(k)}(x_t^{(k)})\|^2] + C_g^2 \beta_t^2 \mathbb{E}[\|\tilde{x}_{t+1}^{(k)} - x_t^{(k)}\|^2] + \frac{\sigma_g^2}{|\mathcal{B}_{t+1}^{(k)}|} \Big),$$

where the last inequality holds due to Assumptions 3 and 4. $\qquad\square$

**Lemma 13.** *In terms of Assumptions 2-4, the following inequality holds.*

$$\sum_{k=1}^{K} \mathbb{E}[\|h_{t+1}^{(k)} - \bar{h}_{t+1}\|^2] \leq \lambda \sum_{k=1}^{K} \mathbb{E}[\|h_t^{(k)} - \bar{h}_t\|^2] + \frac{2C_g^2 L_f^2}{1-\lambda} \sum_{k=1}^{K} \mathbb{E}[\|u_{t+1}^{(k)} - u_t^{(k)}\|^2]$$

$$+ \frac{2\beta_t^2 C_f^2 L_g^2}{1-\lambda} \sum_{k=1}^{K} \mathbb{E}[\|\tilde{x}_{t+1}^{(k)} - x_t^{(k)}\|^2], \tag{60}$$

$$\sum_{k=1}^{K} \mathbb{E}[\|h_0^{(k)} - \bar{h}_0\|^2] \leq \frac{48KC_g^2 L_f^2 \sigma^2}{|\mathcal{B}_0^{(k)}|}.$$

*Proof.* In terms of the definition of $h_{t+1}^{(k)}$, we can get

$$\sum_{k=1}^{K} \mathbb{E}[\|h_{t+1}^{(k)} - \bar{h}_{t+1}\|^2]$$

$$= \mathbb{E}[\|H_{t+1} - \bar{H}_{t+1}\|_F^2]$$

$$= \mathbb{E}[\|H_t W + Q_{t+1} - Q_t - \bar{H}_t - \bar{Q}_{t+1} + \bar{Q}_t\|_F^2]$$

$$\leq (1+a)\mathbb{E}[\|H_t W - \bar{H}_t\|_F^2] + (1+\frac{1}{a})\mathbb{E}[\|Q_{t+1} - Q_t - \bar{Q}_{t+1} + \bar{Q}_t\|_F^2] \tag{61}$$

$$\leq (1+a)\lambda^2 \mathbb{E}[\|H_t - \bar{H}_t\|_F^2] + (1+\frac{1}{a})\mathbb{E}[\|Q_{t+1} - Q_t\|_F^2]$$

$$= \lambda \sum_{k=1}^{K} \mathbb{E}[\|h_t^{(k)} - \bar{h}_t\|^2] + \frac{1}{1-\lambda} \sum_{k=1}^{K} \mathbb{E}[\|q_{t+1}^{(k)} - q_t^{(k)}\|^2],$$

where the last step holds due to $a = \frac{1-\lambda}{\lambda}$. In the following, we will bound the second term in the last inequality.

$$
\begin{aligned}
&\sum_{k=1}^{K} \mathbb{E}[\|q_{t+1}^{(k)} - q_t^{(k)}\|^2] \\
&= \sum_{k=1}^{K} \mathbb{E}[\|\nabla g^{(k)}(x_{t+1}^{(k)})^T \nabla f^{(k)}(u_{t+1}^{(k)}) - \nabla g^{(k)}(x_t^{(k)})^T \nabla f^{(k)}(u_t^{(k)})\|^2] \\
&= \sum_{k=1}^{K} \mathbb{E}[\|\nabla g^{(k)}(x_{t+1}^{(k)})^T \nabla f^{(k)}(u_{t+1}^{(k)}) - \nabla g^{(k)}(x_{t+1}^{(k)})^T \nabla f^{(k)}(u_t^{(k)}) \\
&\quad + \nabla g^{(k)}(x_{t+1}^{(k)})^T \nabla f^{(k)}(u_t^{(k)}) - \nabla g^{(k)}(x_t^{(k)})^T \nabla f^{(k)}(u_t^{(k)})\|^2] \\
&\le \sum_{k=1}^{K} 2\mathbb{E}[\|\nabla g^{(k)}(x_{t+1}^{(k)})^T \nabla f^{(k)}(u_{t+1}^{(k)}) - \nabla g^{(k)}(x_{t+1}^{(k)})^T \nabla f^{(k)}(u_t^{(k)})\|^2] \\
&\quad + \sum_{k=1}^{K} 2\mathbb{E}[\|\nabla g^{(k)}(x_{t+1}^{(k)})^T \nabla f^{(k)}(u_t^{(k)}) - \nabla g^{(k)}(x_t^{(k)})^T \nabla f^{(k)}(u_t^{(k)})\|^2] \\
&\le \sum_{k=1}^{K} 2C_g^2 \mathbb{E}[\|\nabla f^{(k)}(u_{t+1}^{(k)}) - \nabla f^{(k)}(u_t^{(k)})\|^2] + \sum_{k=1}^{K} 2C_f^2 \mathbb{E}[\|\nabla g^{(k)}(x_{t+1}^{(k)}) - \nabla g^{(k)}(x_t^{(k)})\|^2] \\
&\le \sum_{k=1}^{K} 2C_g^2 L_f^2 \mathbb{E}[\|u_{t+1}^{(k)} - u_t^{(k)}\|^2] + \sum_{k=1}^{K} 2\beta_t^2 C_f^2 L_g^2 \mathbb{E}[\|\tilde{x}_{t+1}^{(k)} - x_t^{(k)}\|^2],
\end{aligned}
\tag{62}
$$

where the last step holds due to Assumptions 2-4. By combining above two inequalities, the proof for the first part is completed. When $t = 0$, we can get

$$
\begin{aligned}
&\sum_{k=1}^{K} \mathbb{E}[\|h_0^{(k)} - \bar{h}_0\|^2] \\
&= \sum_{k=1}^{K} \mathbb{E}[\|\nabla g^{(k)}(x_0^{(k)})^T \nabla_g f^{(k)}(u_0^{(k)}) - \frac{1}{K} \sum_{k'=1}^{K} \nabla g^{(k')}(x_0^{(k')})^T \nabla_g f^{(k')}(u_0^{(k')})\|^2] \\
&= \sum_{k=1}^{K} \mathbb{E}[\|\nabla g^{(k)}(x_0^{(k)})^T \nabla_g f^{(k)}(u_0^{(k)}) - \frac{1}{K} \sum_{k'=1}^{K} \nabla g^{(k')}(x_0^{(k')})^T \nabla_g f^{(k)}(u_0^{(k)}) \\
&\quad + \frac{1}{K} \sum_{k'=1}^{K} \nabla g^{(k')}(x_0^{(k')})^T \nabla_g f^{(k)}(u_0^{(k)}) - \frac{1}{K} \sum_{k'=1}^{K} \nabla g^{(k')}(x_0^{(k')})^T \nabla_g f^{(k')}(u_0^{(k')})\|^2] \\
&\le \sum_{k=1}^{K} 2\mathbb{E}[\|\nabla g^{(k)}(x_0^{(k)})^T \nabla_g f^{(k)}(u_0^{(k)}) - \frac{1}{K} \sum_{k'=1}^{K} \nabla g^{(k')}(x_0^{(k')})^T \nabla_g f^{(k)}(u_0^{(k)})\|^2] \\
&\quad + \sum_{k=1}^{K} 2\mathbb{E}[\|\frac{1}{K} \sum_{k'=1}^{K} \nabla g^{(k')}(x_0^{(k')})^T \nabla_g f^{(k)}(u_0^{(k)}) - \frac{1}{K} \sum_{k'=1}^{K} \nabla g^{(k')}(x_0^{(k')})^T \nabla_g f^{(k')}(u_0^{(k')})\|^2] \\
&\le \sum_{k=1}^{K} 2C_f^2 \mathbb{E}[\|\nabla g^{(k)}(x_0^{(k)}) - \frac{1}{K} \sum_{k'=1}^{K} \nabla g^{(k')}(x_0^{(k')})\|^2] + \sum_{k'=1}^{K} 2C_g^2 \mathbb{E}[\|\nabla_g f^{(k)}(u_0^{(k)}) - \frac{1}{K} \sum_{k'=1}^{K} \nabla_g f^{(k')}(u_0^{(k')})\|^2] \\
&\le 8C_g^2 L_f^2 \sum_{k'=1}^{K} \mathbb{E}[\|u_0^{(k)} - \bar{u}_0\|^2] \\
&\le \frac{48K C_g^2 L_f^2 \sigma^2}{|\mathcal{B}_0^{(k)}|}.
\end{aligned}
\tag{63}
$$

$\square$

**Lemma 14.** *In terms of Assumptions 2-4, by setting $|\mathcal{A}_t^{(k)}| = |\mathcal{B}_t^{(k)}| = B$, the following inequality holds.*

$$\sum_{k=1}^{K} \mathbb{E}[\|s_t^{(k)} - \bar{s}_t\|^2] \leq 3 \sum_{k=1}^{K} \mathbb{E}[\|h_t^{(k)} - \bar{h}_t\|^2]$$
$$+ \frac{6K(2C_f^2\sigma_{g'}^2 + 2C_g^2\sigma_f^2)}{B} + \frac{12K(2C_f^2\sigma_{g'}^2 + 2C_g^2\sigma_f^2)}{B}\frac{1}{1-\lambda^2} \ . \tag{64}$$

*Proof.* In terms of the definition of $q_t^{(k)}$, we can get

$$\mathbb{E}[\|z_t^{(k)} - q_t^{(k)}\|^2]$$
$$= \mathbb{E}[\|\nabla g^{(k)}(x_t^{(k)}; \mathcal{B}_t^{(k)})^T \nabla_g f^{(k)}(u_t^{(k)}; \mathcal{A}_t^{(k)}) - \nabla g^{(k)}(x_t^{(k)})^T \nabla_g f^{(k)}(u_t^{(k)})\|^2]$$
$$= \mathbb{E}[\|\nabla g^{(k)}(x_t^{(k)}; \mathcal{B}_t^{(k)})^T \nabla_g f^{(k)}(u_t^{(k)}; \mathcal{A}_t^{(k)}) - \nabla g^{(k)}(x_t^{(k)})^T \nabla_g f^{(k)}(u_t^{(k)}; \mathcal{A}_t^{(k)})$$
$$+ \nabla g^{(k)}(x_t^{(k)})^T \nabla_g f^{(k)}(u_t^{(k)}; \mathcal{A}_t^{(k)}) - \nabla g^{(k)}(x_t^{(k)})^T \nabla_g f^{(k)}(u_t^{(k)})\|^2] \tag{65}$$
$$\leq \frac{2C_f^2\sigma_{g'}^2 + 2C_g^2\sigma_f^2}{B} \ ,$$

where the last inequality holds due to Assumptions 3, 4, and $|\mathcal{A}_t^{(k)}| = |\mathcal{B}_t^{(k)}| = B$. Meanwhile, we can get

$$\sum_{k=1}^{K} \mathbb{E}[\|s_t^{(k)} - \bar{s}_t\|^2]$$
$$= \sum_{k=1}^{K} \mathbb{E}[\|s_t^{(k)} - h_t^{(k)} + h_t^{(k)} - \bar{h}_t + \bar{h}_t - \bar{s}_t\|^2] \tag{66}$$
$$\leq 3 \sum_{k=1}^{K} \mathbb{E}[\|s_t^{(k)} - h_t^{(k)}\|^2] + 3 \sum_{k=1}^{K} \mathbb{E}[\|h_t^{(k)} - \bar{h}_t\|^2] + 3 \sum_{k=1}^{K} \mathbb{E}[\|\bar{h}_t - \bar{s}_t\|^2] \ .$$

In the following, we will bound the first and last terms in the last inequality, respectively.

$$\sum_{k=1}^{K} \mathbb{E}[\|s_{t+1}^{(k)} - h_{t+1}^{(k)}\|^2]$$
$$= \mathbb{E}[\|S_{t+1} - H_{t+1}\|_F^2]$$
$$= \mathbb{E}[\|S_t W + Z_{t+1} - Z_t - H_t W - Q_{t+1} + Q_t\|_F^2]$$
$$= \mathbb{E}[\|Z_{t+1} - Q_{t+1}\|_F^2] + \mathbb{E}[\|(S_t - H_t)W - (Z_t - Q_t)\|_F^2]$$
$$= \mathbb{E}[\|Z_{t+1} - Q_{t+1}\|_F^2] + \mathbb{E}[\|(S_{t-1}W + Z_t - Z_{t-1} - H_{t-1}W - Q_t + Q_{t-1})W - (Z_t - Q_t)\|_F^2]$$
$$= \mathbb{E}[\|Z_{t+1} - Q_{t+1}\|_F^2] + \mathbb{E}[\|(S_{t-1} - H_{t-1})W^2 - (Z_{t-1} - Q_{t-1})W + (Z_t - Q_t)(W - I)\|_F^2]$$
$$= \mathbb{E}[\|Z_{t+1} - Q_{t+1}\|_F^2] + \mathbb{E}[\|(Z_t - Q_t)(W - I)\|_F^2] + \mathbb{E}[\|(S_{t-1} - H_{t-1})W^2 - (Z_{t-1} - Q_{t-1})W\|_F^2]$$
$$= \mathbb{E}[\|Z_{t+1} - Q_{t+1}\|_F^2] + \mathbb{E}[\|(Z_t - Q_t)\|_F^2\|W - I\|_2^2] + \mathbb{E}[\|(S_{t-1} - H_{t-1})W^2 - (Z_{t-1} - Q_{t-1})W\|_F^2]$$
$$= \mathbb{E}[\|Z_{t+1} - Q_{t+1}\|_F^2] + \sum_{j=0}^{t} \mathbb{E}[\|Z_j - Q_j\|_F^2\|W^{t-j}(W - I)\|_2^2]$$
$$\leq \frac{(2C_f^2\sigma_{g'}^2 + 2C_g^2\sigma_f^2)K}{B} + \frac{4K(2C_f^2\sigma_{g'}^2 + 2C_g^2\sigma_f^2)}{B} \sum_{j=0}^{t} \lambda^{2j}$$
$$\leq \frac{K(2C_f^2\sigma_{g'}^2 + 2C_g^2\sigma_f^2)}{B} + \frac{4K(2C_f^2\sigma_{g'}^2 + 2C_g^2\sigma_f^2)}{B}\frac{1}{1-\lambda^2} \ , \tag{67}$$

where the first inequality holds due to Eq. (65), the last inequality holds due to $\lambda < 1$. Additionally, we can get

$$
\sum_{k=1}^{K} \mathbb{E}[\|\bar{h}_t - \bar{s}_t\|^2]
$$

$$
= \sum_{k=1}^{K} \mathbb{E}[\|\frac{1}{K} \sum_{k=1}^{K} \nabla g^{(k)}(x_t^{(k)})^T \nabla_g f^{(k)}(u_t^{(k)}) - \frac{1}{K} \sum_{k=1}^{K} \nabla g^{(k)}(x_t^{(k)}; \mathcal{B}_t^{(k)})^T \nabla_g f^{(k)}(u_t^{(k)}; \mathcal{A}_t^{(k)})\|^2]
$$

$$
\leq \sum_{k=1}^{K} \frac{1}{K} \sum_{k=1}^{K} \mathbb{E}[\|\nabla g^{(k)}(x_t^{(k)})^T \nabla_g f^{(k)}(u_t^{(k)}) - \nabla g^{(k)}(x_t^{(k)}; \mathcal{B}_t^{(k)})^T \nabla_g f^{(k)}(u_t^{(k)}; \mathcal{A}_t^{(k)})\|^2] \tag{68}
$$

$$
= \sum_{k=1}^{K} \frac{1}{K} \sum_{k=1}^{K} \mathbb{E}[\|\nabla g^{(k)}(x_t^{(k)})^T \nabla_g f^{(k)}(u_t^{(k)}) - \nabla g^{(k)}(x_t^{(k)})^T \nabla_g f^{(k)}(u_t^{(k)}; \mathcal{A}_t^{(k)})
$$

$$
+ \nabla g^{(k)}(x_t^{(k)})^T \nabla_g f^{(k)}(u_t^{(k)}; \mathcal{A}_t^{(k)}) - \nabla g^{(k)}(x_t^{(k)}; \mathcal{B}_t^{(k)})^T \nabla_g f^{(k)}(u_t^{(k)}; \mathcal{A}_t^{(k)})\|^2]
$$

$$
\leq \frac{K(2C_g^2 \sigma_f^2 + 2C_f^2 \sigma_{g'}^2)}{B} ,
$$

where the last inequality holds due to Assumptions 3, 4, and $|\mathcal{A}_t^{(k)}| = |\mathcal{B}_t^{(k)}| = B$. By combining above three inequalities, the proof is completed.

$\square$

Similar to the proof of Theorem 1, we set $\beta_t = \beta$, $|\mathcal{A}_t^k| = |\mathcal{B}_t^k| = B$, and introduce a new potential function, which is defined as follows:

$$
P_t = \mathbb{E}[F(\bar{x}_t)] + \frac{4\eta C_g^2 L_f^2}{\gamma} \frac{1}{K} \sum_{k=1}^{K} \mathbb{E}[\|u_t^{(k)} - g^{(k)}(x_t^{(k)})\|^2] + \frac{\eta(1-\lambda)}{\gamma} \frac{1}{K} \sum_{k=1}^{K} \mathbb{E}[\|h_t^{(k)} - \bar{h}_t\|^2]
$$

$$
+ \frac{1}{\gamma} \frac{1}{K} \sum_{k=1}^{K} \mathbb{E}[\|x_t^{(k)} - \bar{x}_t\|^2] . \tag{69}
$$

In terms of these lemmas and definition, we are ready to prove Theorem 2.

*Proof.*

$$P_{t+1} - P_t$$
$$\leq -\frac{\eta\beta}{2}\mathbb{E}[\|\nabla F(\bar{x}_t)\|^2]$$
$$+ \frac{3\eta\beta C_g^2\sigma_f^2}{B} + \frac{3\eta\beta C_f^2\sigma_{g'}^2}{B} + \frac{\beta^2\gamma^2\sigma_g^2}{B}\frac{4\eta C_g^2 L_f^2}{\gamma} + \frac{6\gamma^2\beta^2 C_g^2 L_f^2\sigma_g^2}{(1-\lambda)B}\frac{\eta(1-\lambda)}{\gamma}$$
$$+ \frac{6(2C_f^2\sigma_{g'}^2 + 2C_g^2\sigma_f^2)}{B}\frac{2\beta\eta^2}{1-\lambda^2}\frac{1}{\gamma} + \frac{12(2C_f^2\sigma_{g'}^2 + 2C_g^2\sigma_f^2)}{B}\frac{1}{1-\lambda^2}\frac{2\beta\eta^2}{1-\lambda^2}\frac{1}{\gamma}$$
$$+ \left(3\eta\beta C_g^2 L_f^2 - \gamma\beta\frac{4\eta C_g^2 L_f^2}{\gamma} + \frac{6\gamma^2\beta^2 C_g^2 L_f^2}{1-\lambda}\frac{\eta(1-\lambda)}{\gamma}\right)\frac{1}{K}\sum_{k=1}^{K}\mathbb{E}[\|u_t^{(k)} - g^{(k)}(x_t^{(k)})\|^2]$$
$$+ \left(\eta\beta L_F^2 + \frac{9\beta C_g^2}{\gamma}\frac{4\eta C_g^2 L_f^2}{\gamma} + \left(\frac{2\beta^2 C_f^2 L_g^2}{1-\lambda} + \frac{6\gamma^2\beta^4 C_g^4 L_f^2}{1-\lambda}\right)8\frac{\eta(1-\lambda)}{\gamma}\right.$$
$$\left. + \left(-\beta + \frac{\beta(1+\lambda^2)}{2}\right)\frac{1}{\gamma}\right)\frac{1}{K}\sum_{k=1}^{K}\mathbb{E}[\|x_t^{(k)} - \bar{x}_t\|^2] \tag{70}$$
$$+ \left(\frac{9\eta^2\beta C_g^2}{2\gamma}\frac{4\eta C_g^2 L_f^2}{\gamma} + \left(\frac{2\beta^2 C_f^2 L_g^2}{1-\lambda} + \frac{6\gamma^2\beta^4 C_g^4 L_f^2}{1-\lambda}\right)4\eta^2\frac{\eta(1-\lambda)}{\gamma} - \frac{\eta\beta}{4}\right)\mathbb{E}[\|\bar{s}_t\|^2]$$
$$+ \left(\frac{27\eta^2\beta C_g^2}{2\gamma}\frac{4\eta C_g^2 L_f^2}{\gamma} + \left(\frac{2\beta^2 C_f^2 L_g^2}{1-\lambda} + \frac{6\gamma^2\beta^4 C_g^4 L_f^2}{1-\lambda}\right)12\eta^2\frac{\eta(1-\lambda)}{\gamma}\right.$$
$$\left. + \left((\lambda-1)\frac{\eta(1-\lambda)}{\gamma} + \frac{6\beta\eta^2}{1-\lambda^2}\frac{1}{\gamma}\right)\right)\frac{1}{K}\sum_{k=1}^{K}\mathbb{E}[\|h_t^{(k)} - \bar{h}_t\|^2]$$
$$+ \left(\frac{9\eta^2\beta C_g^2}{2\gamma}\frac{4\eta C_g^2 L_f^2}{\gamma} + \left(\frac{2\beta^2 C_f^2 L_g^2}{1-\lambda} + \frac{6\gamma^2\beta^4 C_g^4 L_f^2}{1-\lambda}\right)4\eta^2\frac{\eta(1-\lambda)}{\gamma}\right)\frac{6(2C_f^2\sigma_{g'}^2 + 2C_g^2\sigma_f^2)}{B}$$
$$+ \left(\frac{9\eta^2\beta C_g^2}{2\gamma}\frac{4\eta C_g^2 L_f^2}{\gamma} + \left(\frac{2\beta^2 C_f^2 L_g^2}{1-\lambda} + \frac{6\gamma^2\beta^4 C_g^4 L_f^2}{1-\lambda}\right)4\eta^2\frac{\eta(1-\lambda)}{\gamma}\right)\frac{12(2C_f^2\sigma_{g'}^2 + 2C_g^2\sigma_f^2)}{B(1-\lambda^2)}.$$

By setting $\beta \leq \min\{\frac{1}{8\gamma}, \frac{1}{2\eta L_F}, 1\}$, $\gamma > 0$, and $\eta \leq \min\{\eta_1, \eta_2, \eta_3\}$, where

$$\eta_1 = \frac{4\gamma(1-\lambda^2)}{8\gamma^2 L_F^2 + 289 C_g^4 L_f^2 + 16 C_f^2 L_g^2}$$
$$\eta_2 = \frac{\gamma}{2\sqrt{19 C_g^4 L_f^2 + C_f^2 L_g^2}} \tag{71}$$
$$\eta_3 = \frac{\sqrt{b^2 + 4ac} - b}{2a}, a = \frac{27 C_g^4 L_f^2}{4\gamma^2} + \frac{3 C_f^2 L_g^2}{8\gamma^2} + \frac{3 C_g^4 L_f^2}{128\gamma^2}, b = \frac{6}{1-\lambda^2}, c = (1-\lambda)^2,$$

we can get

$$P_{t+1} - P_t$$
$$\leq -\frac{\eta\beta}{2}\mathbb{E}[\|\nabla F(\bar{x}_t)\|^2]$$
$$+ \frac{3\eta\beta C_g^2\sigma_f^2}{B} + \frac{3\eta\beta C_f^2\sigma_{g'}^2}{B} + \frac{4\gamma\eta\beta^2 C_g^2 L_f^2\sigma_g^2}{B} + \frac{6\gamma\eta\beta^2 C_g^2 L_f^2\sigma_g^2}{B}$$
$$+ \frac{24\beta\eta^2(C_f^2\sigma_{g'}^2 + C_g^2\sigma_f^2)}{\gamma(1-\lambda^2)B} + \frac{48\beta\eta^2(C_f^2\sigma_{g'}^2 + C_g^2\sigma_f^2)}{\gamma(1-\lambda^2)^2 B} \tag{72}$$
$$+ \left(\frac{18\eta^3\beta C_g^4 L_f^2}{\gamma^2} + \frac{8\eta^3\beta^2 C_f^2 L_g^2}{\gamma} + \frac{24\gamma^2\eta^3\beta^4 C_g^4 L_f^2}{\gamma}\right)\frac{12(C_f^2\sigma_{g'}^2 + C_g^2\sigma_f^2)}{B}$$
$$+ \left(\frac{18\eta^3\beta C_g^4 L_f^2}{\gamma^2} + \frac{8\eta^3\beta^2 C_f^2 L_g^2}{\gamma} + \frac{24\gamma^2\eta^3\beta^4 C_g^4 L_f^2}{\gamma}\right)\frac{24(C_f^2\sigma_{g'}^2 + C_g^2\sigma_f^2)}{B(1-\lambda^2)}.$$

By summing over $t$ from 0 to $T - 1$, we can get

$$\frac{1}{T}\sum_{t=0}^{T-1}\frac{\eta\beta}{2}\mathbb{E}[\|\nabla F(\bar{x}_t)\|^2]$$

$$\leq \frac{P_0 - P_T}{T} + \frac{3\eta\beta C_g^2\sigma_f^2}{B} + \frac{3\eta\beta C_f^2\sigma_{g'}^2}{B} + \frac{4\gamma\eta\beta^2 C_g^2 L_f^2\sigma_g^2}{B} + \frac{6\gamma\eta\beta^2 C_g^2 L_f^2\sigma_g^2}{B}$$

$$+ \frac{24\beta\eta^2(C_f^2\sigma_{g'}^2 + C_g^2\sigma_f^2)}{\gamma(1-\lambda^2)B} + \frac{48\beta\eta^2(C_f^2\sigma_{g'}^2 + C_g^2\sigma_f^2)}{\gamma(1-\lambda^2)^2 B}$$

$$+ \Big(\frac{18\eta^3\beta C_g^4 L_f^2}{\gamma^2} + \frac{8\eta^3\beta^2 C_f^2 L_g^2}{\gamma} + \frac{24\gamma^2\eta^3\beta^4 C_g^4 L_f^2}{\gamma}\Big)\frac{12(C_f^2\sigma_{g'}^2 + C_g^2\sigma_f^2)}{B}$$

$$+ \Big(\frac{18\eta^3\beta C_g^4 L_f^2}{\gamma^2} + \frac{8\eta^3\beta^2 C_f^2 L_g^2}{\gamma} + \frac{24\gamma^2\eta^3\beta^4 C_g^4 L_f^2}{\gamma}\Big)\frac{24(C_f^2\sigma_{g'}^2 + C_g^2\sigma_f^2)}{B(1-\lambda^2)} \qquad (73)$$

$$\leq \frac{\mathbb{E}[F(x_0) - F(x_*)]}{T} + \frac{3\eta\beta C_g^2\sigma_f^2}{B} + \frac{3\eta\beta C_f^2\sigma_{g'}^2}{B} + \frac{4\gamma\eta\beta^2 C_g^2 L_f^2\sigma_g^2}{B} + \frac{6\gamma\eta\beta^2 C_g^2 L_f^2\sigma_g^2}{B}$$

$$+ \frac{24\beta\eta^2(C_f^2\sigma_{g'}^2 + C_g^2\sigma_f^2)}{\gamma(1-\lambda^2)B} + \frac{48\beta\eta^2(C_f^2\sigma_{g'}^2 + C_g^2\sigma_f^2)}{\gamma(1-\lambda^2)^2 B} + \frac{4\eta C_g^2 L_f^2\sigma_g^2 + 48\eta(1-\lambda)C_g^2 L_f^2\sigma^2}{\gamma B}$$

$$+ \Big(\frac{18\eta^3\beta C_g^4 L_f^2}{\gamma^2} + \frac{8\eta^3\beta^2 C_f^2 L_g^2}{\gamma} + \frac{24\gamma^2\eta^3\beta^4 C_g^4 L_f^2}{\gamma}\Big)\frac{12(C_f^2\sigma_{g'}^2 + C_g^2\sigma_f^2)}{B}$$

$$+ \Big(\frac{18\eta^3\beta C_g^4 L_f^2}{\gamma^2} + \frac{8\eta^3\beta^2 C_f^2 L_g^2}{\gamma} + \frac{24\gamma^2\eta^3\beta^4 C_g^4 L_f^2}{\gamma}\Big)\frac{24(C_f^2\sigma_{g'}^2 + C_g^2\sigma_f^2)}{B(1-\lambda^2)}$$

$$+ \frac{4\eta C_g^2 L_f^2\sigma_g^2 + 48\eta(1-\lambda)C_g^2 L_f^2\sigma^2}{\gamma B}\,,$$

where the last step holds due to $P_T > F(x_*)$ and

$$P_0 = \mathbb{E}[F(\bar{x}_0)] + \frac{4\eta C_g^2 L_f^2}{\gamma}\frac{1}{K}\sum_{k=1}^{K}\mathbb{E}[\|u_0^{(k)} - g^{(k)}(x_0^{(k)})\|^2] + \frac{\eta(1-\lambda)}{\gamma}\frac{1}{K}\sum_{k=1}^{K}\mathbb{E}[\|h_0^{(k)} - \bar{h}_0\|^2]$$

$$+ \frac{1}{\gamma}\frac{1}{K}\sum_{k=1}^{K}\mathbb{E}[\|x_0^{(k)} - \bar{x}_0\|^2] \qquad (74)$$

$$\leq \mathbb{E}[F(\bar{x}_0)] + \frac{4\eta C_g^2 L_f^2\sigma_g^2 + 48\eta(1-\lambda)C_g^2 L_f^2\sigma^2}{\gamma B}\,.$$

By dividing $\frac{\eta\beta}{2}$ on both sides of this inequality, we can get

$$\frac{1}{T}\sum_{t=0}^{T-1}\mathbb{E}[\|\nabla F(\bar{x}_t)\|^2]$$

$$\leq \frac{2\mathbb{E}[F(x_0) - F(x_*)]}{\eta\beta T} + \frac{6C_g^2\sigma_f^2}{B} + \frac{6C_f^2\sigma_{g'}^2}{B} + \frac{8\gamma\beta C_g^2 L_f^2\sigma_g^2}{B} + \frac{12\gamma\beta C_g^2 L_f^2\sigma_g^2}{B}$$

$$+ \frac{48\eta(C_f^2\sigma_{g'}^2 + C_g^2\sigma_f^2)}{\gamma(1-\lambda^2)B} + \frac{96\eta(C_f^2\sigma_{g'}^2 + C_g^2\sigma_f^2)}{\gamma(1-\lambda^2)^2 B} + \frac{8C_g^2 L_f^2\sigma_g^2 + 96(1-\lambda)C_g^2 L_f^2\sigma^2}{\gamma\beta B}$$

$$+ \Big(\frac{18\eta^2 C_g^4 L_f^2}{\gamma^2} + \frac{8\eta^2\beta C_f^2 L_g^2}{\gamma} + \frac{24\gamma^2\eta^2\beta^3 C_g^4 L_f^2}{\gamma}\Big)\frac{24(C_f^2\sigma_{g'}^2 + C_g^2\sigma_f^2)}{B}$$

$$+ \Big(\frac{18\eta^2 C_g^4 L_f^2}{\gamma^2} + \frac{8\eta^2\beta C_f^2 L_g^2}{\gamma} + \frac{24\gamma^2\eta^2\beta^3 C_g^4 L_f^2}{\gamma}\Big)\frac{48(C_f^2\sigma_{g'}^2 + C_g^2\sigma_f^2)}{B(1-\lambda^2)} \qquad (75)$$

$$\leq \frac{2\mathbb{E}[F(x_0) - F(x_*)]}{\eta\beta T} + \frac{6C_g^2\sigma_f^2 + 6C_f^2\sigma_{g'}^2 + 3C_g^2 L_f^2\sigma_g^2}{B}$$

$$+ \frac{24\eta^2(C_f^2\sigma_{g'}^2 + C_g^2\sigma_f^2)(19C_g^4 L_f^2 + C_f^2 L_g^2)}{\gamma^2 B} + \frac{48\eta^2(C_f^2\sigma_{g'}^2 + C_g^2\sigma_f^2)(19C_g^4 L_f^2 + C_f^2 L_g^2)}{\gamma^2(1-\lambda^2)B}$$

$$+ \frac{48\eta(C_f^2\sigma_{g'}^2 + C_g^2\sigma_f^2)}{\gamma(1-\lambda^2)B} + \frac{96\eta(C_f^2\sigma_{g'}^2 + C_g^2\sigma_f^2)}{\gamma(1-\lambda^2)^2 B} + \frac{8C_g^2 L_f^2\sigma_g^2 + 96(1-\lambda)C_g^2 L_f^2\sigma^2}{\gamma\beta B}\,,$$

where the last inequality holds due to $\beta \leq \frac{1}{8\gamma}$.

$\square$