# OpenReview forum: "Fast Training  Method for  Stochastic Compositional Optimization Problems"
_NeurIPS.cc/2021/Conference — NeurIPS 2021 Poster_

### Official Review · Reviewer_AZDJ · 2021-07-16

**Rating:** 7
**Confidence:** 2

**Summary:**

The paper presents two algorithms for distributed optimization of stochastic composite objectives where one is interested in optimizing one outer objective that is computed on points that depend on another function (called the inner objective in the paper).
The authors adapt two strategies for distributed optimization to the stochastic composite optimization problem and derive convergence rates for the general non-convex outer objective problems, that show linear rates w.r.t. the number of devices used (this is probably the main contribution of this paper).
Finally, they present a set of experiments in meta-learning using MAML as meta-learning algorithm.

**Main Review:**

Clarity:
The paper is well written and generally clear. Few exceptions are:
- missing definition of the non-stochastic versions of g and f, used, for instance, in Eq. (2). I interpret these to be $\mathbb{E} f$ and g, but this should be clarified.
- perhaps, more importantly, I find a discussion on the limit case where K=1 (i.e. only one device) missing. How do the algorithms read in this case? and, what are the similarities/differences with [14]?
- one case where composite functions emerge in machine learning is bilvel optimization (not limited to the MAML setting). In bilevel optimization typically the term ''inner objective''/inner function is reserved to the (scalar-valued) objective of the inner problem. In this work, $g$ essentially maps to the argmin operator, which is in general not necessarily a function (could be multiple minimizes). I think it would be appropriate to include a brief comment on this after the authors introduce the main assumptions, perhaps also linking to some recent works on stochastic bilevel optimization, e.g. [1].

Originality and significance:
The work seems original and significant to me, as the methods proposed may be applied to many problems of interest in ML, especially in meta-learning. However, I am not particularly an expert in the field of distributed optimization, so cannot evaluate properly this point.

Quality:
The derivations seem correct to me, although I have not checked carefully the supplementary material.
- One main concern that I have about this work is that I think there is a disconnect between the theoretical results and the experimental section. Specifically, as the authors claim that one main contribution of the paper is about showing linear rates w.r.t. the number of devices, I would have expected some empirical results in this regard. Instead, all the experiments are conducted with only 4 devices, which makes it impossible to appreciate this aspect of the paper. This point should be address in the rebuttal, ideally with some empirical result.

Typos:
- L80 (the)
- L88 (of)

[1] Grazzi, Riccardo, Massimiliano Pontil, and Saverio Salzo. "Convergence properties of stochastic hypergradients." International Conference on Artificial Intelligence and Statistics. PMLR, 2021.

____
Post-rebuttal:

Thanks for your reply! I am raising my score and suggest acceptance.

**Time Spent Reviewing:**

5

---

> ### Author Response · Authors · 2021-08-09
> **Response to Reviewer AZDJ**
>
> We are grateful to the review for recognizing the contributions of our work.
>
> * Thanks for pointing out that unclear part. The definition should be
>
>      $g(x)= \mathbb{E}_{\xi}[g(x; \xi)]$
>
>      $f(y)= \mathbb{E}_{\zeta}[f(y; \zeta)]$.
>
> * To achieve $\epsilon$-accuracy solution, the number of iterations $T$ is $O(\frac{1}{K\epsilon^2})$, and the batch size is $B=O(\sqrt{KT})=O(\frac{1}{\epsilon})$. Thus, the computation complexity is $T\times B= O(\frac{1}{K\epsilon^3})$. Our computation complexity has two advantages over $O(\frac{1}{\epsilon^4})$ of the single-machine SCGD [14]. On the one hand, by setting $K=1$, our computation complexity is better than the traditional SCGD, which is a significant improvement over SCGD. On the other hand, by setting $K>1$, our computation complexity is further improved due to the collaboration of multiple devices. We will give more discussion about this important contribution.
>
> *  Thanks for your kind suggestion. In fact, there do exist some relationships between the bilevel optimization problem and the stochastic compositional optimization problem. For instance, the meta-learning problem is a bilevel optimization problem. As shown in Section 5.1, it can be reformulated as a stochastic compositional optimization problem. We will cite the related work that you mentioned and give more discussions about the relationship between the GENERAL bilevel optimization problem and the stochastic compositional optimization problem in our final version.
>
> * As you suggested, we verify the performance of our algorithms with different devices. Due to the space limitation, we show the result of GT-DSCGD for the regression task in the following table. We will include other results in our final version. In Columns 2-3, we show the consumed time and the objective value when using 4 devices. In Columns 4-5, we show those items when using 8 devices. It can be observed that the consumed time of 4 devices is almost doubled compared with that of 8 devices, which can confirm the speedup of our parallel algorithm.
>
> | Iteration | Seconds(4 devices) | Loss (4 devices) | Loss (8 devices) | Seconds (8 devices) |   |
> |-----------|--------------------|------------------|------------------|---------------------|---|
> | 10        | 1.6                | 4.18             | 3.96             | 1.0                 |   |
> | 100       | 185                | 2.56             | 2.52             | 113                 |   |
> | 200       | 421                | 1.72             | 2.05             | 228                 |   |
> | 300       | 621                | 1.03             | 1.33             | 372                 |   |
> | 400       | 823                | 0.93             | 1.04             | 476                 |   |
> | 500       | 1038               | 0.73             | 0.79             | 580                 |   |
> | 1000      | 2094               | 0.58             | 0.51             | 1172                |   |
>
>
>
> * We will carefully refine our writing in the final version.

---

### Official Review · Reviewer_HU5v · 2021-07-20

**Rating:** 6
**Confidence:** 5

**Summary:**

This paper studies the stochastic compositional optimization problem under the distributed learning setting. Two decentralized optimization methods are proposed for solving this problem, both of which are based on SCGD method. Theoretical analysis is conducted to provide convergence bound for both methods. Experiments are also carried out accordingly.

**Limitations And Societal Impact:**

N/A.

**Main Review:**

After rebuttal:
The authors' clarification about the overall complexity of the algorithm seems reasonable to me. I decided to raise the score. Hope these discussions about the algorithm complexity can be combined into the final version.

=====================================================

My main concern about this paper is that the theoretical bounds are not well discussed and compared with existing methods.

It is only mentioned in the paper that the number of iterations required by the proposed methods is $T=O(1/(K\epsilon^2))$, the same as decentralized SGD. But according to Corollaries 1 and 3, the proposed methods use the batch size $B=O(\sqrt{KT})$, which depends on the number of nodes $K$ and the number of iterations $T$, while decentralized SGD only needs batch size 1. So, it is unfair to directly compare the number of iterations. The overall computation complexity, which takes both factors into account, should be discussed and compared with decentralized SGD and the single-machine SCGD to see if the proposed methods really have linear speedup and optimal convergence rate.

The values of the hyperparameters used in the experiments are not mentioned. Especially for the batch size, is the theoretically optimal batch size $B=O(\sqrt{KT})$ used here, or just set $B=1$? Also, the batch size of the baseline should also be clarified.

Besides, I guess there are some typos or errors in the convergence analysis part. What is $\beta$? Is it $\beta_t$ is the algorithm? I also failed to locate the term $1/(2\eta L_F)$ mentioned in Remark 2.

One more suggestion: since this paper focus on distributed optimization, it is better to reflect this in the title (though I am not sure if the title can be modified in this stage).

**Time Spent Reviewing:**

4

---

> ### Author Response · Authors · 2021-08-09
> **Response to Reviewer HU5v**
>
> We are grateful to the review for recognizing the contributions of our work.
>
> * To achieve the $\epsilon$-accuracy solution, i.e., $\frac{1}{T}\sum_{t=0}^{T-1}\mathbb{E}[||\nabla F(\bar{{x}}_{t})||^2]\leq \epsilon$,  the number of iterations $T$ is $O(\frac{1}{K\epsilon^2})$, and the batch size is $B=O(\sqrt{KT})=O(\frac{1}{\epsilon})$. Thus, the computation complexity is $T\times B= O(\frac{1}{K\epsilon^3})$. Our computation complexity has two advantages over $O(\frac{1}{\epsilon^4})$ of the single-machine SCGD [14]. On the one hand, by setting $K=1$, our computation complexity $O(\frac{1}{\epsilon^3})$  is better than $O(\frac{1}{\epsilon^4})$ of the traditional SCGD, which is a significant improvement over SCGD. We will give more discussion about this contribution in our final version. On the other hand, by setting $K>1$, our computation complexity is further improved due to the collaboration of multiple devices. At last, the sample complexity $O(\frac{1}{K\epsilon^3})$ also indicates that our algorithm can achieve the linear speedup with respect to the number of workers $K$. In summary, the computation complexity of our algorithm is much better than the traditional SCGD.
>
> * The computation complexity of the traditional SGD method is $O(1/\epsilon^2)$, while that of SCGD is $O(1/\epsilon^4)$, which is caused by the complicated structure of the stochastic compositional optimization problem. As a result, this intrinsic property of SCGD causes that decentralized SGD has a better computation complexity $O(\frac{1}{K\epsilon^2})$  than $O(\frac{1}{K\epsilon^3})$ of decentralized SCGD. However, we would like to emphasize that our method has the same communication complexity $O(\frac{1}{K\epsilon^2})$   with decentralized SGD. Moreover, in our experiments, we use the same mini-batch size for both decentralized SGD and decentralized SCGD. Thus, their PRACTICAL computation complexities are in the same order, but our method demonstrates a faster convergence speed in Figure 1.
>
> * For the regression task, we use a fixed batch size 200 on each device for all methods, which is stated in Line 239. For the classification task, we set it to 8 on each device for all methods, which is stated in Line 267. We will make these settings clearer in our final version.
>
> * In our algorithm, we set $\beta=\beta_t$. $\eta$ is the learning rate, which is shown in Line 10 of Algorithm 1 and Line 11 of Algorithm 2. $L_F$ is defined in Line 167.
>
> * Thanks for your suggestion. We will try to revise the title if allowed.
>
> * At last, we would like to emphasize the contributions of our work. First, our work is the first one studying decentralized SCGD and establishing its convergence rate. It is novel in both theoretical analysis and practical applications. Second, when setting $K=1$, our algorithm can achieve a better computation complexity than traditional SCGD, which is a significant improvement over traditional SCGD. Thus, our work also has important contributions for the single machine SCGD method. In summary, our work is novel and has important contributions for the development of SCGD.

---

### Official Review · Reviewer_eCMs · 2021-07-21

**Rating:** 6
**Confidence:** 3

**Summary:**

This paper proposed two decentralized training methods for stochastic compositional optimization problem and establish the convergence results for them.


**Limitations And Societal Impact:**

The authors have properly addressed them.

**Main Review:**

This paper introduces the decentralized training methods to solve the stochastic compositional optimization. Although I don’t get into the whole proof, the techniques used in this paper are fairly standard and the analysis is technically sound. My main concern is the significance of this paper.

Minor Comments:

1. I suggest the authors replace the ‘fast training methods’ in the title with e.g.’decentralized training’ as this is a paper on the decentralized training, rather than accelerating the convergence (on single machine).

2. Please formalize the definition of $g$ and $f$. Note that the $f$ and $g$ in (1) depends on some random variable, but are deterministic functions in (2) and some parts hereafter.

3. Should it be better to denote the difference of $f^{(k)}$ and $g^{(k)}$ in (4) by the difference of the corresponding random variables? This means the difference are induced by different batch of data that are sampled according to the random variables, which avoid some potential ambiguity. Note that in (3) the authors use some similar notation. I feel the current notations mean that each device have different $f$ and $g$.

4. I would like to ask what’s the relation of the analysis in this paper and analysis of smooth SCGD and smooth distributed SGD. It would be better to illustrate the technical significance beyond the direct combination of the previous work. Specifically, given that the gradient variance are bounded, it’s not so hard to derive Lemma 14 that replace $s_t$ with $h_t$ as emphasized in Line 201-209. It would be more interesting if there’s other technical insight in this paper.

5. For experiments, I think the convergence rate with different $K$ is necessary, as the authors argue it as a main contribution.

I tend to accept this paper, but due to the concern on the technical significance, I would rather assign a conservative score for it.


**Time Spent Reviewing:**

2

---

> ### Author Response · Authors · 2021-08-09
> **Response to Reviewer eCMs**
>
> We are grateful to the review for recognizing the contributions of our work.
>
> * Thanks for your kind suggestion. We will try to revise the title in our final version.
>
> * Thanks for pointing out that unclear part. The definition should be
>
>     $g(x)=\mathbb{E}_{\xi}[g(x; \xi)]$,
>
>     $f(y)=\mathbb{E}_{\zeta}[f(y;\zeta)]$
>
> * Yes, we will refine the definition of $f^{(k)}$ and $g^{(k)}$ as you suggested to make it clear in our final version.
>
> * Compared with the single machine SCGD algorithm [14] and the distributed SGD algorithm, our algorithm is much more challenging. On the one hand, to establish the convergence rate, a critical step is to bound the consensus error, which is much more difficult than existing algorithms due to the complicated compositional gradient. We successfully addressed this challenge and established the convergence rate for both the gossip-based algorithm and the gradient-tracking-based algorithm. On the other hand, we would like to emphasize the importance of introducing $h_t^{(k)}$ in Eq. (21). If directly using $s_t^{(t)}$ rather than $h_t^{(k)}$, there will be an additive term that is not divided by the batch size or the number of iterations. Then, the final convergence bound will have an additive term so that it can only show the proposed algorithm converges to the neighborhood of the stationary point. In our proof, by introducing $h_t^{(k)}$, we remove that additive term. Then, by setting $B=O(1/\epsilon)$, we can prove that our algorithm can converge to the stationary point. Thus, introducing $h_t^{(k)}$ is critical for establishing the convergence rate of Algorithm 2. At last, we would like to further emphasize that our algorithm can achieve better computation complexities, which is a significant improvement over the single machine SCGD algorithm. In particular, to achieve the $\epsilon$-accuracy solution, i.e., $\frac{1}{T}\sum_{t=0}^{T-1}\mathbb{E}[||\nabla F(\bar{{x}}_{t})||^2]\leq \epsilon$, the  computation complexity of our algorithm is $O(\frac{1}{K\epsilon^3})$. When $K=1$ (the single-machine setting), our computation complexity is $O(\frac{1}{\epsilon^3})$, which is much better than $O(\frac{1}{\epsilon^4})$ of the traditional SCGD algorithm [14]. Thus, the contribution of our algorithm is significant.
>
> * As you suggested, we verify the performance of our algorithms with different devices. Due to the space limitation, we show the result of GT-DSCGD for the regression task in the following table. We will include other results in our final version. In Columns 2-3, we show the consumed time and the objective value when using 4 devices. In Columns 4-5, we show those items when using 8 devices. It can be observed that the consumed time of 4 devices is almost doubled compared with that of 8 devices, which can confirm the speedup of our parallel algorithm.
>
> | Iteration | Seconds(4 devices) | Loss (4 devices) | Loss (8 devices) | Seconds (8 devices) |   |
> |-----------|--------------------|------------------|------------------|---------------------|---|
> | 10        | 1.6                | 4.18             | 3.96             | 1.0                 |   |
> | 100       | 185                | 2.56             | 2.52             | 113                 |   |
> | 200       | 421                | 1.72             | 2.05             | 228                 |   |
> | 300       | 621                | 1.03             | 1.33             | 372                 |   |
> | 400       | 823                | 0.93             | 1.04             | 476                 |   |
> | 500       | 1038               | 0.73             | 0.79             | 580                 |   |
> | 1000      | 2094               | 0.58             | 0.51             | 1172                |   |

---

### Official Review · Reviewer_P4ej · 2021-07-23

**Rating:** 6
**Confidence:** 4

**Summary:**

This paper studies decentralized stochastic compositional optimization problem, and proposes two algorithms: GP-DSCGD and GT-DSCGD. They also have convergence guarantee for both algorithms. This problem has significant application in model agnostic meta learning.


**Limitations And Societal Impact:**

If my understanding is correct, the gradient tracking based algorithm has the same convergence rate with GP-DSCGD, then I was left wondering what would be its significance.

Also, I notice that we need a fairly large batch size $\sqrt{KT}$ to guarantee the convergence and appreciate clarification from authors.  In particular, is it due to the decentralized setting, or the single machine version of the algorithm also needs a large batch size?


**Main Review:**

The problem studied in this paper has significant application in large scale distributed meta learning, so I think this work has nice results except issues discussed below. As far as I know, this paper is the first to study decentralized stochastic compositional optimization, and provide analysis. From my point of view, it is novel. The theoretical analysis looks sound as far as I checked. The theoretical results are supported through experiments. The paper is well-organized and both experimental and theoretical results are clearly presented.


**Time Spent Reviewing:**

3

---

> ### Author Response · Authors · 2021-08-09
> **Response to Reviewer P4ej**
>
> We are grateful to the review for recognizing the contributions of our work.
>
> * To achieve the $\epsilon$-accuracy solution, i.e., $\frac{1}{T}\sum_{t=0}^{T-1}\mathbb{E}[||\nabla F(\bar{{x}}_{t})||^2]\leq \epsilon$, the number of iterations $T$ is in the order of $O(\frac{1}{K\epsilon^2})$, and then the batch size is $B=O(\sqrt{KT})=O(\frac{1}{\epsilon})$. Here, a large batch size is helpful to reduce the variance. Existing non-parallel stochastic compositional gradient descent algorithms, such as [1-2], also use a large batch size to reduce the gradient variance for accelerating the convergence speed.  Note that, in the empirical evaluation, we can just use a fixed number of batch size. For instance, the meta-batch size of the regression task is set to 200 in our experiment.
>
> * The gradient-tracking-based algorithm has the same THEORETICAL convergence rate as the gossip-based algorithm, which is confirmed by existing works [3]. However, when different devices have different data distributions, the EMPIRICAL performance of the gradient-tracking-based algorithm could be better than the gossip-based algorithm. The reason is that the gradient-tracking-based algorithm communicates both model parameters and gradients, while the gossip-based algorithm only communicates model parameters. Thus, the heterogeneous gradient of the gossip-based algorithm may degenerate the convergence performance. We will add the simulation experiment to further show the performance of the gradient-tracking-based algorithm in our final version.
>
>
> [1]. Yuan, Huizhuo, Xiangru Lian, and Ji Liu. "Stochastic Recursive Variance Reduction for Efficient Smooth Non-Convex Compositional Optimization." arXiv preprint arXiv:1912.13515 (2019).
>
> [2]. Chen, Ziyi, and Yi Zhou. "Momentum with variance reduction for nonconvex composition optimization." arXiv preprint arXiv:2005.07755 (2020).
>
> [3]. Pu, Shi, and Angelia Nedić. "A distributed stochastic gradient tracking method." 2018 IEEE Conference on Decision and Control (CDC). IEEE, 2018.

---

> > ### Comment · Area_Chair_2ugh · 2021-08-18
> > **followup question**
> >
> > It seems that you only count the number of iterations and the communication cost. How about the sample complexity? More and more samples are required to estimate the expectation inside. It is much higher than the iteration complexity. The sample complexity result is not consistent with SCGD. Can authors explain it?

---

> > > ### Author Response · Authors · 2021-08-18
> > > **Sample complexity**
> > >
> > > Thank you for your great question. We answer your question in the following.
> > >
> > > To achieve the $\epsilon$-accuracy solution, i.e., $\frac{1}{T}\sum_{t=0}^{T-1}\mathbb{E}[||\nabla F(\bar{{x}}_{t})||^2]\leq \epsilon$,  the number of iterations $T$ is $O(\frac{1}{K\epsilon^2})$, and the batch size is $B=O(\sqrt{KT})=O(\frac{1}{\epsilon})$. Thus, the computation /sample complexity is $T\times B= O(\frac{1}{K\epsilon^3})$. Our sample complexity has two advantages over $O(\frac{1}{\epsilon^4})$ of the single-machine SCGD [14]. On the one hand, by setting $K=1$, our sample complexity $O(\frac{1}{\epsilon^3})$  is better than $O(\frac{1}{\epsilon^4})$ of the traditional SCGD, which is a significant improvement over SCGD. We will give more discussions about this contribution in our final version. On the other hand, by setting $K>1$, our sample complexity is further improved due to the collaboration of multiple devices.  In summary, the sample complexity of our algorithm is much better than the traditional SCGD.

---

> > > > ### Comment · Area_Chair_2ugh · 2021-08-19
> > > > **Can you compare the result of [15]?**
> > > >
> > > > The sample complexity in [15] is $O(1/\epsilon^{2.25})$ or equivalently the convergence rate (wrt the sample complexity) is $O(1/T^{4/9})$ where T is the number of samples?

---

> > > > > ### Author Response · Authors · 2021-08-20
> > > > > **Compare the sample complexity with that of [15]**
> > > > >
> > > > > The algorithm proposed in [15] used the **acceleration** technique to accelerate the convergence speed. In particular, it introduces an additional variable $a_t$ (Here, we use the symbols in our paper) and computes $u_{t+1}$ as follows:
> > > > >
> > > > > $a_{t+1}= (1-\frac{1}{b_t})x_{t} + \frac{1}{b_t}x_{t+1}$
> > > > >
> > > > > $u_{t+1} = (1-b_t)u_{t}+ b_t g(\textcolor{red}{a_{t+1}})$
> > > > >
> > > > > On the contrary, our method does not use the acceleration technique and computes $u_{t+1}$ by directly using $x_{t+1}$:
> > > > >
> > > > > $u_{t+1} = (1-b_t)u_{t}+ b_t g(\textcolor{red}{x_{t+1}})$
> > > > >
> > > > > This is the main difference between our method and [15] when setting $K=1$.
> > > > >
> > > > > Due to the employed  **acceleration** technique in [15], its sample complexity $O(1/\epsilon^{2.25})$ is better than $O(1/\epsilon^3)$ of our method.
> > > > >
> > > > > We thank AC points out the difference in the sample complexity between our method and [15]. We will include more discussions regarding [15] in our final version. Additionally, our decentralized training framework can be extended to the accelerated variant in [15]. We will leave it as our future work and further improve the sample complexity of [15] with our techniques.
> > > > >
> > > > > At last, we would like to emphasize that the goal and main contribution of our work are developing the **decentralized** stochastic compositional gradient descent method to efficiently train the stochastic compositional optimization problem. To the best of our knowledge, this is the first work studying this problem and providing theoretical guarantees. Thus, our work is novel and has important contributions to SCGD.
> > > > >
> > > > > We hope that we have answered your question.

---

### Decision · Program_Chairs · 2021-09-27

**Decision:**

Accept (Poster)

**Comment:**

This paper considers the Stochastic Compositional Optimization on the decentralized training setting, which is an interesting and novel problem, as recognized by reviewers. A decentralized stochastic compositional gradient method is proposed. The proved convergence rate is provided as well. Most reviewers' comments have been addressed. Although we decide to accept this paper, authors should realize that this rate is not tight, comparing to the special case (single machine), which should be explicitly pointed out in the revision.